# Molecular basis for the folding of β-helical autotransporter passenger domains

Xiaojun Yuan[1], Matthew D. Johnson[1], Jing Zhang[1], Alvin W. Lo[2,3], Mark A. Schembri [2,3], Lakshmi C. Wijeyewickrema[4], Robert N. Pike[4], Gerard H.M. Huysmans[5], Ian R. Henderson[6] & Denisse L. Leyton[1,7]

Bacterial autotransporters comprise a C-terminal β-barrel domain, which must be correctly folded and inserted into the outer membrane to facilitate translocation of the N-terminal passenger domain to the cell exterior. Once at the surface, the passenger domains of most autotransporters are folded into an elongated β-helix. In a cellular context, key molecules catalyze the assembly of the autotransporter β-barrel domain. However, how the passenger domain folds into its functional form is poorly understood. Here we use mutational analysis on the autotransporter Pet to show that the β-hairpin structure of the fifth extracellular loop of the β-barrel domain has a crucial role for passenger domain folding into a β-helix. Bioinformatics and structural analyses, and mutagenesis of a homologous autotransporter, suggest that this function is conserved among autotransporter proteins with β-helical passenger domains. We propose that the autotransporter β-barrel domain is a folding vector that nucleates folding of the passenger domain.

[1] Research School of Biology, Australian National University, Canberra, ACT 0200, Australia. [2] School of Chemistry and Molecular Biosciences, The University of Queensland, Brisbane, QLD 4072, Australia. [3] Australian Infectious Disease Research Centre, The University of Queensland, Brisbane, QLD 4072, Australia. [4] Department of Biochemistry and Genetics, La Trobe Institute for Molecular Science, La Trobe University, Melbourne, VIC 3086, Australia. [5] Department of Physiology and Biophysics, Weill Cornell Medicine, New York, NY 10065, USA. [6] Institute of Microbiology and Infection, University of Birmingham, Birmingham B15 2TT, UK. [7] Medical School, Australian National University, Canberra, ACT 0200, Australia. These authors contributed equally: Xiaojun Yuan, Matthew D. Johnson. Correspondence and requests for materials should be addressed to D.L.L. (email: denisse.leyton@anu.edu.au)

Type V autotransporter proteins (currently classified as types Va to Ve) are a distinct class of outer membrane proteins (OMPs) that share a common secretion pathway, which facilitates their navigation through the bacterial cell envelope[1]. Although the different subtypes of type V autotransporters are distinguished by variations in their domain organization, oligomerization state, and structure, they are unified by a similar transmembrane topology[1]. Embedded in the outer membrane is a β-barrel domain that forms a secretion pore, which is needed for the translocation of the (often covalently linked) passenger domain to the bacterial cell surface[1].

The prototypical type Va ("classical") autotransporter consists of a 12-stranded C-terminal β-barrel domain and an N-terminal passenger domain[2] that, in most cases, is folded or predicted to fold into an elongated β-helix[3–6], although globular folds have also been reported[7] and predicted to occur[6]. Once folded, passenger domains have key roles in pathogenesis; some passenger domains remain attached to the β-barrel to function as adhesins, whereas others are processed and released into the environment to function as toxins, or to mediate disruption of the host immune response[8–11].

In a cellular context, key molecules catalyze classical autotransporter folding and membrane insertion as they do for other types of β-barrel proteins. The translocation and assembly module participates in autotransporter assembly[12–14], as do periplasmic chaperones such as Skp and SurA[15–18]. Based on evidence from cross-linking studies that show autotransporters directly interact with BamA, the central component of the β-barrel assembly machinery[15,19,20], it has been proposed that the passenger domain is translocated through a pore formed by the transient fusion of the autotransporter and BamA β-barrels during membrane insertion. BamA structures and molecular dynamics simulations show conformational flexibility along the seam that hydrogen bonds the first with the last β-strand of the BamA β-barrel[21,22]. However, it is unclear whether BamA can transiently form fusion pores with substrate OMPs.

Irrespective of the composition of the translocation pore, there is a general agreement that in classical autotransporters, the passenger domain is translocated in a C- to- N-terminal direction where the most C-terminal β-helical rungs fold into a stable, protease-resistant structure called the 'stable core'[3,19,23,24]. It has been proposed that folding of the portion of the passenger domain that appears first at the cell surface into a stable scaffold nucleates processive folding of the more N-terminal segments of the passenger domain during translocation. In support of this hypothesis are chemical[3,24,25] and mechanical[26] denaturation studies showing that the core is the most stable part of the protein. Moreover, several lines of evidence suggest that stacking (π–π) interactions between buried aromatic residues in the stable core not only provide stability to this region, but also drive translocation and vectorial folding of the passenger domain[19,26,27]. Although collectively these results provide insight into how the passenger domain is translocated and folded into its functional form, many mechanistic details are lacking.

Here, using the classical Pet autotransporter, an enterotoxin from enteroaggregative *Escherichia coli*[28] as a model protein, we present evidence that the fifth β-hairpin loop (L5), which connects β-strands 9 and 10 at the extracellular surface of the β-barrel domain, templates folding of the passenger domain. By monitoring the assembly of Pet variants carrying modifications in the L5 β-hairpin in live bacterial cells, we provide evidence that β-strand propensity of the loop, but not its specific amino acid sequence is required for folding of the passenger domain into a stable β-helix. Moreover, biophysical studies on refolded protein suggest that the nucleation of passenger domain folding can occur independently of the cellular machinery, and is entirely mediated by the β-barrel domain. Experimental analysis of a second classical autotransporter, EspP, and in silico sequence and structure comparison with other autotransporters lead us to suggest that L5-assisted folding of the passenger domain represents a conserved folding mechanism for classical autotransporter proteins.

## Results

**Folded Pet is largely resistant to proteinase K digestion**. The homologous autotransporter EspP from enterohaemorrhagic *E. coli* has been crystallized[29,30], thus providing a means to model the topology of the Pet β-barrel domain (Fig. 1a) and to infer its three-dimensional structure (Fig. 1b). Pet and EspP belong to the SPATE (serine protease autotransporters of *Enterobacteriaceae*) subfamily of autotransporters, which are characterized by an autocatalytic processing of the passenger from the β-barrel between two conserved asparagine residues in the intra-barrel α-helical segment that connects both domains[30–32] (Fig. 1a, b). Processing of the full-length ~ 136 kDa Pet and EspP pro-proteins results in the secretion of the ~ 106 kDa mature form of the passenger domains into the culture supernatant, whereas the ~ 30 kDa β-barrel domains remain embedded in the outer membrane[28,33] (Fig. 1c). We have established a time-course assay for the pulse-chase expression of Pet in whole cells that allows its assembly to be monitored by SDS-polyacrylamide gel electrophoresis (PAGE) and immunoblotting using antibodies that selectively recognize the Pet passenger domain[34]. Moreover, the addition of proteinase K to a duplicate sample at each time-point provides information about the state of folding of the molecule[34]. Indeed, while periplasmic proteins are protected from proteinase K digestion, unless the outer membrane is first permeabilized by polymyxin B[34] (Supplementary Fig. 1a, b), the passenger is partially digested into a ~ 70 kDa protease-resistant fragment when surface exposed and natively folded[34] (Fig. 1e, left panel, + PK).

**Truncation of extracellular L5 perturbs Pet assembly in vivo**. Truncation of a long β-hairpin loop (L4) that connects β-strands 7 and 8 at the extracellular surface of the β-barrel domain of BrkA, a classical autotransporter from *Bordetella pertussis*, was shown to reduce the extent of passenger domain translocation[35]. Hence, we wanted to test whether the longer extracellular loops of the β-barrel domain (Fig. 1a, b, green) also have a role in the assembly of SPATEs. Thus, we truncated the loops that connect β-strands 9 and 10 (L5; PetΔL5) (Fig. 1d), 7 and 8 (L4; PetΔL4), and 5 and 6 (L3; PetΔL3) (Supplementary Fig. 2a), and compared the assembly of these mutant proteins with the assembly of wild type Pet using the pulse-chase expression assay[34]. Consistent with previous results[34], processing of full-length Pet (which here refers to Pet minus its signal peptide) into distinct passenger and β-barrel fragments was in process by 5 min and largely completed by 10 min (Fig. 1e, left panel, – PK). Proteinase K digestion of Pet into a ~ 70 kDa protease-resistant fragment on the same timescale showed that the mature passenger domain molecules were natively folded (Fig. 1e, left panel, + PK). In contrast, the ~ 106 kDa mature PetΔL5 passenger was absent at all time points (Fig. 1e, right panel, ± PK). Furthermore, although full-length PetΔL5 was detected by the 5 min time point (Fig. 1e, right panel, – PK), complete digestion of the passenger domain molecules by proteinase K was observed (Fig. 1e, right panel, + PK), suggesting that they were surface exposed in an unfolded or severely misfolded conformation. The presence of cleaved and similarly heat-modifiable Pet and PetΔL5 β-barrels in total membrane preparations, detected using antibodies that selectively recognize the Pet β-barrel domain (Fig. 1g), suggests that truncation of L5 does not inhibit folding or insertion of the β-barrel into bacterial outer membranes, or autocatalytic release of the passenger

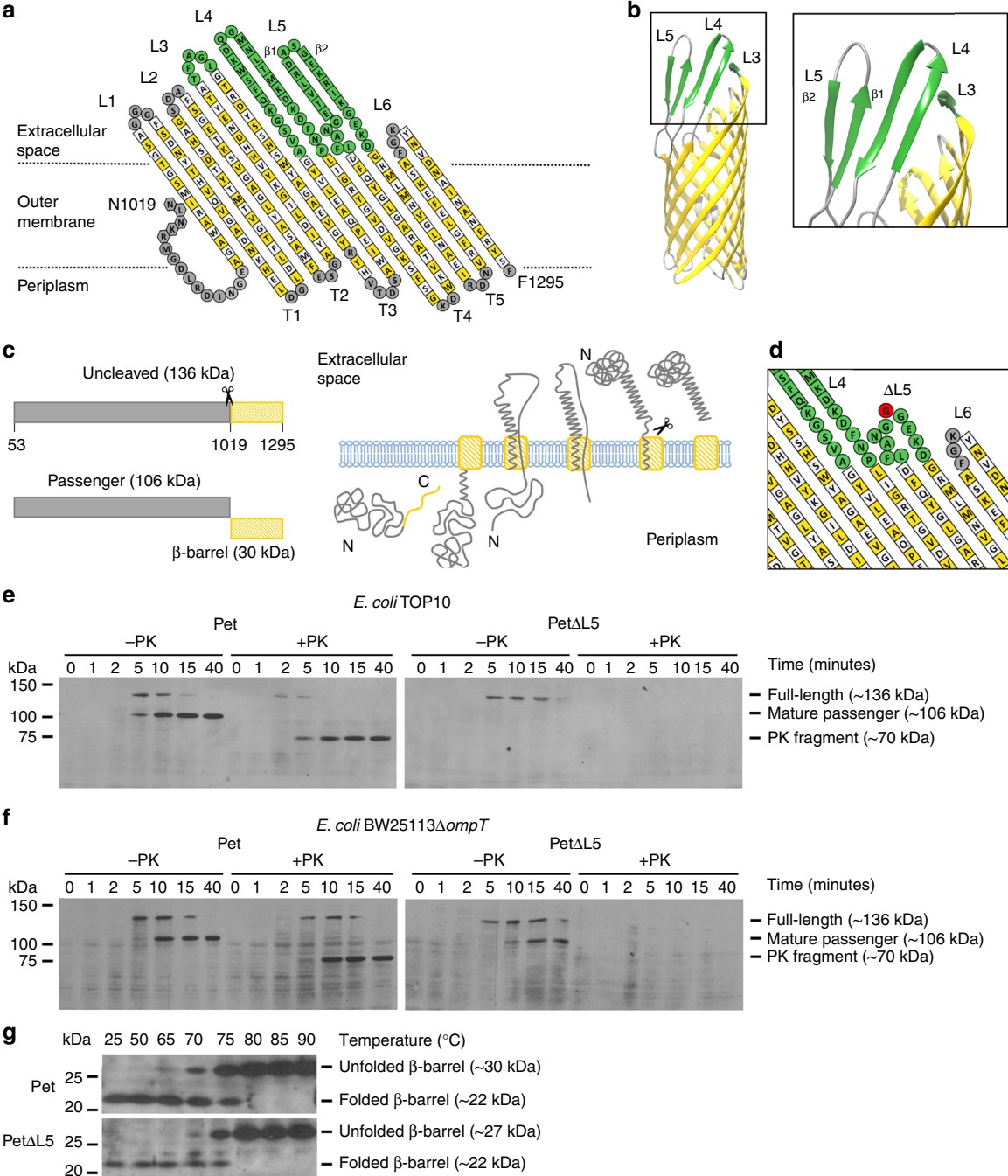

**Fig. 1** Truncation of L5 affects folding of the Pet passenger domain in bacterial cells. Topology model of the Pet β-barrel domain (**a**) based on the crystal structure of the EspP β-barrel domain (PDB code 3SLJ) (these β-barrels share ~ 91% sequence identity) (**b**). The structure shows L5 in an 'open' conformation where it interacts with L4, forming a 4-stranded β-sheet. It is noteworthy that the uncleaved passenger domain that traverses the length of the pore is hidden for clarity. For **a**, yellow and white squares indicate residues with side chains that point toward the lipid bilayer and the β-barrel lumen, respectively. Residues in periplasmic turns labeled T1–T5 and extracellular loops labeled L1, L2, and L6 are shown in gray, as are the residues in the α-helical segment. Residues in extracellular loops labeled L3, L4, and L5 are shown in green. Given that these loops are mobile, the alignment of L4 and L5 does not strictly follow PDB 3SLJ. **c** The domain structure (left) and schematic for biogenesis (right) show that, post cytoplasmic synthesis and inner membrane translocation, Pet is processed by a signal peptidase to a ~ 136 kDa polypeptide in the periplasm. The β-barrel domain of Pet is then assembled into the outer membrane to allow translocation of the passenger domain, in a C- to- N- terminal direction, until the extreme N-terminus traverses the β-barrel pore. An autocatalytic cleavage reaction within an intra-barrel α-helical segment liberates the ~ 106 kDa mature form of the passenger domain, leaving the ~ 30 kDa β-barrel domain embedded in the outer membrane. **d** Topology model of the Pet β-barrel domain showing the L5 truncation created by replacing $E_{1233}$–$K_{1246}$ with one glycine residue (shown in red). Pulse-chase expression of Pet and PetΔL5 and sensitivity to proteinase K (PK) in *E. coli* TOP10 (**e**) and BW25113Δ*ompT* (**f**) monitored by SDS-PAGE and immunoblotting with anti-Pet passenger domain antibodies. All samples were TCA precipitated prior to SDS-PAGE. **g** Heating of total membranes containing Pet and PetΔL5 at the temperatures indicated. Samples were analyzed by SDS-PAGE and immunoblotting with anti-β-barrel domain antibodies. Images are representative of at least two independent experiments

domain into the culture supernatant. These data are consistent with a previous study on EspP where Barnard et al.[29] aimed to investigate the contribution of L5 on β-barrel domain stability. In addition to finding that truncation of L5 in EspP ($\Delta E_{1238}$–$E_{1253}$) had no effect on the stability of this domain, the authors also noted no effect on passenger domain translocation or cleavage.

To account for the disappearance of the ~ 106 kDa mature PetΔL5 passenger, pulse-chase expression of Pet and PetΔL5 was repeated in E. coli BW25113$\Delta ompT$[36], a strain lacking the outer membrane protease OmpT. OmpT digests unfolded passenger domains on the bacterial surface, while natively folded passengers are resistant to OmpT degradation[37]. As expected, pulse-chase expression assays showed that the assembly of Pet in E. coli TOP10 (Fig. 1e, left panel, ± PK), BW25113$\Delta ompT$ (Fig. 1f, left panel, ± PK), and wild-type BW25113 (Supplementary Fig. 3, left panel, ± PK) occurred in a similar fashion. The assembly of PetΔL5 in TOP10 (Fig. 1e, right panel, – PK) and BW25113 in the absence of proteinase K (Supplementary Fig. 3, right panel, – PK) also occurred in a similar manner. In contrast, there were clear differences in the assembly of PetΔL5 in BW25113$\Delta ompT$ in the absence of proteinase K (Fig. 1f, right panel, – PK). In BW25113$\Delta ompT$, full-length PetΔL5 was visible by the 5 min time point where processing into distinct passenger and β-barrel fragments occurred in a manner similar to wild type Pet (Fig. 1f, left panel, – PK). These data suggest that in TOP10 and BW25113, the PetΔL5 passenger is autocatalytically cleaved from its β-barrel domain, yet is unable to fold into its native conformation and is therefore susceptible to degradation by OmpT. Moreover, the presence of the full-length ~ 136 kDa species in all three strains suggests that when attached to its β-barrel, the passenger domain is less accessible to OmpT proteolysis (Fig. 1e, f and Supplementary Fig. 3, right panels, – PK). Nevertheless, complete degradation of the ~ 136 kDa and ~ 106 kDa species observed after proteinase K treatment (Fig. 1e, f and Supplementary Fig. 3, right panels, + PK) demonstrates that the PetΔL5 passenger, whether in its full-length form and still attached to the L5-truncated β-barrel or cleaved and released into the culture supernatant as a mature protein, is unable to fold into its native conformation when exposed on the bacterial surface. Barnard et al.[29] did not report a similar defect in passenger folding upon L5-truncation in EspP. However, the EspP variant used by these authors [EspPΔ1(ΔL5)] only contained the 116 residues of the passenger domain immediately N-terminal to the L5-truncated β-barrel domain where the small passenger size (~ 13 kDa) would make it challenging to accurately pick up folding defects using pulse-chase assays[29].

There was no major difference in the folding of PetΔL3 relative to that observed in wild type Pet, where the generation of a ~ 70 kDa protease-resistant fragment in samples treated with proteinase K suggested that the passenger domain was able to fold into its native conformation (Supplementary Fig. 2b, + PK). In contrast, the cleavage of PetΔL4 occurred with substantially reduced efficiency, with very little processing of the 136 kDa species to the 106 kDa mature species after 40 min (Supplementary Fig. 2b, – PK). The 136 kDa form of PetΔL4 was only degraded by proteinase K in the presence of polymyxin B, suggesting that most of the passenger is exposed to the periplasm (Supplementary Fig. 2c, + polymyxin B and PK). However, the low translocation efficiency of PetΔL4 makes it difficult to assess whether L4 also has a role in passenger folding (Supplementary Fig. 2b, + PK). Pulse-chase expression of PetΔL4 in BW25113$\Delta ompT$ visualized greater amounts of the 136 kDa and 106 kDa species (Supplementary Fig. 2d, e, – PK), suggesting that some of the translocated PetΔL4 passenger domain is susceptible to OmpT degradation and is thus unfolded. The low translocation efficiency of PetΔL4 was rescued in a mutant with a partially

truncated L4 β-hairpin, PetΔL4P (Supplementary Fig. 2f). PetΔL4P displayed a concomitant delay in the arrival of the passenger domain at the cell surface and autocatalytic processing into distinct passenger and β-barrel domain fragments relative to that observed in wild type Pet (Supplementary Fig. 2g, ± PK). Generation of a ~ 70 kDa protease-resistant fragment in samples treated with proteinase K showed that the passenger domain of PetΔL4P was able to fold into its native conformation despite the delay in its surface exposure (Supplementary Fig. 2g, + PK). Collectively, these data indicate that the L4 β-hairpin is mainly required for passenger translocation to occur efficiently, as shown previously for a L4 truncation variant of BrkA[35]. Furthermore, while L4 could have a role in passenger folding, either directly or indirectly, it appears not to be essential for this process.

**Passenger folding relies on the β-hairpin conformation of L5.** To identify the amino acid residues of L5 responsible for mediating passenger folding, Pet variants were created (Fig. 2a) and analyzed by pulse-chase expression. Pet$^{L5\beta1/G}$, a variant with all amino acids in the first β-strand in L5 ($E_{1233}$–$D_{1238}$) mutated to glycine, demonstrated a severe defect in passenger domain folding similar to that observed for PetΔL5 (Fig. 2c). As a small portion of the 136 kDa form of Pet$^{L5\beta1/G}$ was not degraded by proteinase K (Fig. 2c, + PK), we performed pulse-chase expression of Pet$^{L5\beta1/G}$ in BW25113$\Delta ompT$ to test whether a translocation defect is the explanation for the absence of the ~ 70 kDa proteinase-resistant fragment that is a marker for correct passenger folding. In BW25113$\Delta ompT$, full-length Pet$^{L5\beta1/G}$ was visible by the 5 min time point where processing into distinct passenger and β-barrel fragments occurred in a manner similar to wild-type Pet (Supplementary Fig. 4a, b, – PK), whereas treatment with proteinase K failed to yield a prominent ~ 70 kDa protease-resistant fragment (Supplementary Fig. 4b, + PK). These data confirm that in TOP10, the Pet$^{L5\beta1/G}$ passenger is autocatalytically cleaved from its β-barrel domain, yet is unable to efficiently fold into its native conformation and is therefore susceptible to degradation by OmpT. In contrast, there were no obvious differences in the folding of Pet$^{L5\beta2/G}$, a variant with all amino acids in the second β-strand in L5 ($E_{1242}$–$K_{1246}$) mutated to glycine, relative to that observed in wild type Pet (Fig. 2c). Pet$^{L5Un/G}$, a variant with all amino acids in the unstructured region beneath the β-hairpin ($L_{1228}$–$N_{1231}$ and $E_{1248}$–$D_{1250}$) mutated to glycine, was similarly unaffected (Fig. 2c). Together, these data suggest that a stretch of six amino acids ($E_{1233}$–$D_{1238}$) in the first β-strand in L5 mediate folding of the Pet passenger domain.

To determine whether the side-chains of the six residues in first β-strand in L5 facilitate passenger folding by providing specificity to the interaction, a series of site-directed mutants were created (Supplementary Fig. 5a) and analyzed by pulse-chase expression. There were no obvious differences in the folding of mutants in which all hydrophobic residues were mutated to the polar amino acid asparagine (Pet$^{L5VLI/N}$) or in which charged residues were swapped to the opposite charge (Pet$^{L5R/D,E/K}$) to disrupt potential salt bridges with the second β-strand (Supplementary Fig. 5c), relative to that observed in wild type Pet (Supplementary Fig. 5b). Finally, there was no obvious difference in the folding of Pet$^{L5TD/A}$, a mutant with the two remaining residues in the first β-strand in L5 ($T_{1234}$ and $D_{1238}$) mutated to alanine, relative to that observed in wild-type Pet (Supplementary Fig. 5a, b). The tolerance of amino acids with dissimilar physicochemical properties in all of these positions suggests that the nucleation of passenger folding does not depend on specific amino acid side-chain interactions. Rather, passenger folding might still occur, because the substitutions do not disrupt backbone–backbone

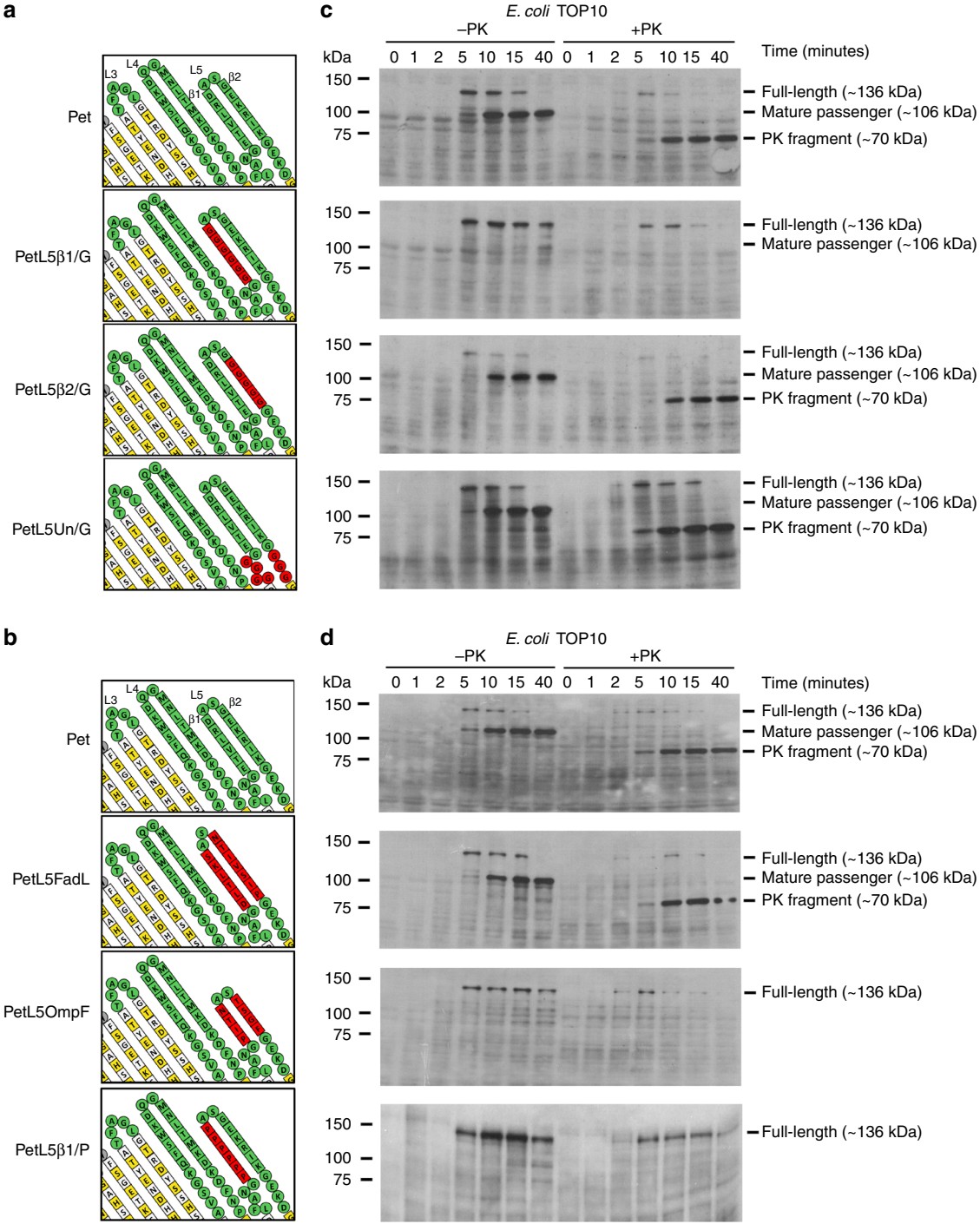

**Fig. 2** L5 likely templates folding of the Pet passenger domain by β-strand augmentation. Topology model of the Pet β-barrel domain showing the Pet$^{L5β1/G}$, Pet$^{L5β2/G}$, and Pet$^{L5Un/G}$ (**a**), and the Pet$^{L5FadL}$, Pet$^{L5OmpF}$, and Pet$^{L5β1/P}$ (**b**) mutations (in red). Pulse-chase expression of Pet, Pet$^{L5β1/G}$, Pet$^{L5β2/G}$, and Pet$^{L5Un/G}$ (**c**), and of Pet, Pet$^{L5FadL}$, Pet$^{L5OmpF}$, and Pet$^{L5β1/P}$ (**d**), and sensitivity to proteinase K (PK) in E. coli TOP10 monitored by SDS-PAGE and immunoblotting with anti-Pet passenger domain antibodies. All samples were TCA precipitated prior to SDS-PAGE. Images are representative of at least two independent experiments

hydrogen bond interactions between the passenger and L5, nor the β-sheet structure of the L5 β-hairpin. This hypothesis is consistent with our findings demonstrating that substitution of the residues in the first β-strand in L5 with a stretch of six glycine residues severely perturbs folding of the passenger domain (Fig. 2a, c) conceivably through disruption of the β-sheet structure in this region (the average solution conformation of a six residue glycine-based peptide is a $3_1$ extended helix[38,39]). To

further validate whether disruption of the β-sheet structure perturbs passenger folding, we substituted all amino acids in the first β-strand in L5 for proline to create Pet$^{L5β1/P}$, eliminating potential hydrogen bonding partners required for the formation of a β-sheet (Fig. 2b). Similar to Pet$^{L5β1/G}$, Pet$^{L5β1/P}$ had a severe defect in passenger folding (Fig. 2d), strongly supporting the hypothesis that β-strand structure in L5 is required for efficient passenger folding. As in Pet$^{L5β1/G}$, a small portion of the 136 kDa

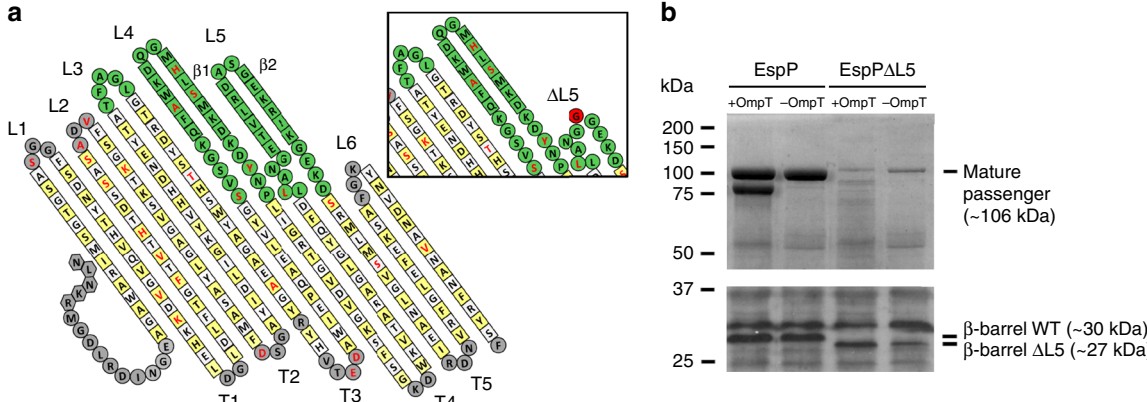

**Fig. 3** Truncation of L5 affects folding of the EspP passenger domain. **a** Left panel: Topology model of the EspP β-barrel domain based on its crystal structure (PDB code 3SLJ). Yellow and white squares indicate residues with side chains that point toward the lipid bilayer and the β-barrel lumen, respectively. Residues in periplasmic turns labeled T1-T5 and extracellular loops labeled L1, L2, and L6 are shown in gray, as are the residues in the α-helical segment. Residues in extracellular loops labeled L3, L4, and L5 are shown in green and residues that differ in Pet are highlighted in red font. Right panel: topology model of the EspP β-barrel domain showing the L5 truncation created by replacing $E_{1238}$–$K_{1251}$ with one glycine residue (shown in red). **b** SDS-PAGE of TCA-precipitated culture supernatant fractions (top panel) and immunoblotting of whole-cell lysates with anti-Pet β-barrel domain antibodies* (bottom panel) collected after growth of *E. coli* TOP10 ( + OmpT) and BW25113Δ*ompT* (− OmpT) expressing EspP and EspPΔL5. Image is representative of at least two independent experiments. *The Pet and EspP β-barrel domains are so similar that anti-Pet β-barrel domain antibodies cross-react with the EspP β-barrel domain

form of $Pet^{L5\beta1/P}$ was not degraded by proteinase K (Fig. 2d). Therefore, we performed pulse-chase expression of $Pet^{L5\beta1/P}$ in BW25113Δ*ompT*, which again ruled out a translocation defect as an explanation for the absence of a prominent ~ 70 kDa proteinase-resistant fragment (Supplementary Fig. 4a, b, ± PK).

Next, we replaced the residues $E_{1233}$–$D_{1238}$ and $G_{1241}$–$K_{1246}$ in L5 with residues $D_{316}$–$S_{321}$ and $N_{324}$–$P_{330}$ from loop 5 from the long-chain fatty acid transporter FadL, where similar to L5 in Pet the first β-strand in the β-hairpin is six residues in length, and with residues $P_{239}$–$N_{242}$ and $T_{247}$–$F_{250}$ from loop 6 from the porin OmpF, which has a shorter four-residue β-strand 1 to create $Pet^{L5FadL}$ and $Pet^{L5OmpF}$, respectively (Fig. 2b and Supplementary Fig. 5d). There was no obvious difference in the folding of $Pet^{L5FadL}$ relative to that observed in wild type Pet (Fig. 2d). In contrast, pulse-chase expression of $Pet^{L5OmpF}$ demonstrated a severe defect in passenger domain folding similar to that observed for PetΔL5 (Fig. 2d). Yet again, pulse-chase expression of $Pet^{L5OmpF}$ in BW25113Δ*ompT* (Supplementary Fig. 4a and b, ± PK) confirmed that, in TOP10, the mature $Pet^{L5OmpF}$ passenger domain is largely unfolded and therefore susceptible to degradation by OmpT. One explanation for this discrepancy is that the heterologous OmpF loop does not adopt a β-hairpin when grafted onto the Pet β-barrel domain. However, if the heterologous OmpF loop does adopt a β-hairpin conformation when grafted onto the Pet β-barrel domain, then these data suggest that β-strand propensity and the length of the L5 β-hairpin, but not its specific amino acid sequence, have a role in the efficient folding of the Pet passenger domain.

**L5-assisted folding could be a conserved folding mechanism.** To test whether L5-assisted folding of the passenger domain is specific to Pet, we examined the influence of the L5 truncation on the SPATE EspP[29,30] by truncating the same amino acid residues in L5 of EspP as in Pet to create EspPΔL5 (Fig. 3a). The Pet and EspP β-barrel domains are 91% identical (Supplementary Fig. 6b) and have L5 regions that differ by only one amino acid in 23 residues, where a $F_{1229}$ to L substitution is found in the unstructured region beneath the β-hairpin (Fig. 3a). In contrast,

although both passenger domains fold into an elongated β-helix[40,41], their sequence identity is 50%. Hence, EspP is ideally suited to investigate whether L5-assisted folding of the passenger domain could represent a conserved folding mechanism. Owing to the absence of available anti-EspP passenger antibodies for pulse-chase analysis, culture supernatant fractions from TOP10 and BW25113Δ*ompT* cells expressing EspP and EspPΔL5 were analyzed for the presence of cleaved passenger domain. SDS-PAGE demonstrated that the EspP passenger secreted from BW25113Δ*ompT* accumulated in the culture supernatant as one distinct fragment, whereas a breakdown product was observed when secreted from TOP10 (Fig. 3b, top panel). This breakdown product has been observed previously when EspP was over-expressed from a high-copy plasmid in an *ompT* + *E. coli* strain[42]. In contrast, SDS-PAGE demonstrated that the EspPΔL5 passenger secreted from both strains accumulated in the culture supernatant at negligible levels, despite the presence of cleaved EspPΔL5 β-barrels in whole cell lysates (Fig. 3b, bottom panel). Together, these data suggest that the L5 β-hairpin also mediates folding of the EspP passenger domain.

Although the conservation of residues within L5 of Pet and EspP is consistent with a conserved mechanism of passenger domain folding, the question becomes why is the sequence so conserved if it can be varied? To address this question, we performed a multiple sequence alignment using Muscle[43] on the amino acid sequence of SPATE and non-SPATE β-barrel domains that are representative of 11 distinct Types of β-barrel domains[6] (Supplementary Fig. 6a). These distinct β-barrel Types are categorized based on the arrangement of conserved motifs and phylogenetic clustering[6]. Identity between SPATE β-barrel domains, which are all Type 5 β-barrel domains, is high, ranging from 59.21% to 99.64% (Supplementary Fig. 6b). From this, it follows that L5 is the same length in all SPATE β-barrel domains and contains several residues that are highly or completely conserved. In contrast, residues within L5 of non-SPATE β-barrel domains differ substantially with each other and with those in SPATE β-barrel domains (Supplementary Fig. 6a). Percent identity between these β-barrel domains ranges from 11.64% to 37.46% and even those belonging to the same β-barrel domain

type share low sequence identity (Supplementary Fig. 6c). Together, these data suggest that the conservation of residues within L5 in SPATE β-barrel domains could be reflective of the high conservation of the β-barrel domains among the SPATE subfamily and vice versa for non-SPATE β-barrel domains.

Next, we examined structures of solved non-SPATE β-barrel domains with electron density in the L5 region to determine whether L5 regions with low sequence identity to Pet or EspP still form a β-hairpin. This analysis revealed that the L5 region in the EstA[7] and AIDA-I[44] β-barrels both form a β-hairpin (Supplementary Fig. 7a, b). Although there is no structure for the AIDA-I passenger domain, it is predicted to be structurally homologous to that of Ag43, a β-helical passenger domain that mediates biofilm formation in *E. coli* through self-association and cell aggregation[45–47]. In contrast, the EstA passenger domain has a globular fold where a central four-stranded parallel β-sheet sits in close proximity to the L5 β-hairpin[7] (Supplementary Fig. 7a). To test whether the EstA and AIDA-I β-hairpins have the potential to promote folding, we replaced the residues in the Pet L5 β-hairpin with residues $D_{556}$–$L_{561}$ and $L_{564}$–$E_{572}$ from loop 5 from EstA, and with residues $G_{1227}$–$L_{1237}$ from loop 5 of AIDA-I to create Pet$^{L5EstA}$ and Pet$^{L5AIDA-I}$, respectively (Supplementary Fig. 7c). Similar to L5 in Pet, the first β-strand in the EstA β-hairpin is six residues in length, whereas that of the AIDA-I β-hairpin is shorter and only four residues in length. Pulse-chase expression assays showed that there was no obvious difference in the folding of Pet$^{L5EstA}$ relative to that observed in wild type Pet (Supplementary Fig. 7d). When grafted onto the Pet β-barrel domain, the short AIDA-I β-hairpin was also able to support folding of the Pet passenger domain. Although it is unclear why the first β-strand in the AIDA-I β-hairpin, but not the OmpF β-hairpin, is able to promote folding of the Pet passenger domain, despite both β-strands being four residues in length, it supports the notion that β-strand propensity and length might play a role in passenger folding. Nevertheless, we can conclude that while sequence alignments provide little insight into whether L5-assisted passenger folding might be conserved, studies with grafted heterologous loops coupled with structure-based analysis make way for a conserved role for L5 in the folding of SPATE and non-SPATE passenger domains, regardless of whether the passenger fold is β-helical or globular with mixed α/β-content.

**Passenger folding occurs without cellular machinery in vitro.** To investigate whether L5 of Pet is critical for folding of the passenger domain independently of the cellular machinery, the folding of Pet was monitored directly and in isolation. To this end, we engineered plasmids for the overexpression of proteins Pet$^{\Delta 1-554}$ and PetΔL5$^{\Delta 1-554}$, comprising the wild-type and L5-truncated β-barrel, respectively, and the 464 residues (~ 51 kDa) of the passenger located immediately N-terminal to the β-barrel domain (Fig. 4a). Inclusion bodies containing Pet$^{\Delta 1-554}$ and PetΔL5$^{\Delta 1-554}$ were solubilized in urea and refolded in vitro by rapid dilution of the denaturant into detergent micelles as described previously[34]. The efficiency of the refolding reaction was monitored by the appearance of ~ 30 kDa and ~ 51 kDa cleavage products corresponding to the β-barrel and released passenger domains, respectively, from autocatalytic processing of the full-length ~ 81 kDa proteins (Fig. 4b).

To compare the folding of the Pet$^{\Delta 1-554}$ and PetΔL5$^{\Delta 1-554}$ passenger domains, a time course trypsin assay was used to generate unique tryptic fingerprints caused by the transient exposure of trypsin-accessible arginine and lysine residues during passenger folding. In this assay, aliquots of the reaction mixture, at increasing incubation times, were treated with trypsin and then

mixed with SDS to stop folding prior to SDS-PAGE analysis and immunoblotting using anti-passenger antibody (Fig. 4c, top panel). The folded Pet$^{\Delta 1-554}$ passenger was largely resistant to trypsinolysis, generating trypsin-resistant fragments between ~ 35 and ~ 40 kDa, whereas the PetΔL5$^{\Delta 1-554}$ passenger was degraded completely. The presence of cleaved β-barrels in duplicate trypsin treated samples, detected using anti-β-barrel antibody (Fig. 4c, bottom panel), indicated that truncation of L5 does not inhibit autocatalytic release of the passenger into the refolding buffer. Thus, we purified the Pet$^{\Delta 1-554}$ and PetΔL5$^{\Delta 1-554}$ passengers isolated from the refolding buffer (Supplementary Fig. 8a) and characterized their fold using spectroscopic methods. The far-UV circular dichroism (CD) spectrum demonstrated that only the passenger derived from Pet$^{\Delta 1-554}$ acquired secondary structure after refolding, whereas the spectrum of the PetΔL5$^{\Delta 1-554}$ passenger appeared almost indistinguishable from that of the unfolded protein in guanidine hydrochloride (Fig. 4d). Similarly, the tryptophan fluorescence spectrum of the passenger derived from Pet$^{\Delta 1-554}$ demonstrated a maximum emission wavelength ($\lambda_{max}$) of 321 nm, whereas the spectrum of the PetΔL5$^{\Delta 1-554}$ passenger appeared almost indistinguishable from that of the unfolded protein in guanidine hydrochloride, with a red-shifted $\lambda_{max}$ of 331 nm (Fig. 4e). Together, these data suggest that the Pet β-barrel is a folding vector where L5 assists folding possibly by providing a structural template for the nucleation of passenger folding, a process that occurs independently of the cellular machinery in vitro.

**Passenger misfolding is not caused by a misfolded β-barrel.** As autocatalytic processing of SPATE passengers requires a folded β-barrel domain[29,31,32], the presence of cleaved Pet$^{\Delta 1-554}$ and PetΔL5$^{\Delta 1-554}$ β-barrels in trypsin-treated samples (Fig. 4c, bottom panel) also suggested that truncation of L5 does not inhibit folding of this domain in vitro. To validate these findings, we analyzed the structure of the refolded, cleaved and purified Pet$^{\Delta 1-554}$ and PetΔL5$^{\Delta 1-554}$ β-barrel domains (Supplementary Fig. 8b) by CD spectroscopy. CD spectra of both the Pet$^{\Delta 1-554}$ and PetΔL5$^{\Delta 1-554}$ β-barrels showed a characteristic β-signature with a minimum at 218 nm (Fig. 4f). Furthermore, CD collected as a function of temperature where denaturation of β-sheet structure is monitored in 1% SDS at 218 nm, showed that the apparent thermal stability of the PetΔL5$^{\Delta 1-554}$ β-barrel is indistinguishable from that of Pet$^{\Delta 1-554}$ (Fig. 4g). Together, these data confirm that both β-barrel domains are folded correctly and strongly suggest that the folding defect observed for the passenger domain derived from PetΔL5$^{\Delta 1-554}$ is a direct consequence of the L5-truncation.

## Discussion

How autotransporter passenger domains fold into a β-helix in the absence of a chemical energy source or electrochemical gradient has remained controversial. In this study, we analyzed the folding of Pet in bacterial cells and in vitro. We find that the first β-strand in L5 is required to efficiently and rapidly fold the passenger domain into its native, protease-resistant structure, suggesting that it acts as a scaffold for passenger folding in a manner that is reminiscent of β-strand augmentation. β-strand augmentation is a process where non-covalent protein–protein interactions in a variety of biological processes are mediated by the addition of a β-strand in an unstructured ligand to a β-strand or β-sheet in a preformed, independently folded receptor, resulting in the formation of a continuous β-sheet[48]. For example, β-strand augmentation is one of the proposed mechanisms by which BamA mediates folding of OMP substrates[49–51]. Furthermore, the observation that the length and β-strand propensity of the very

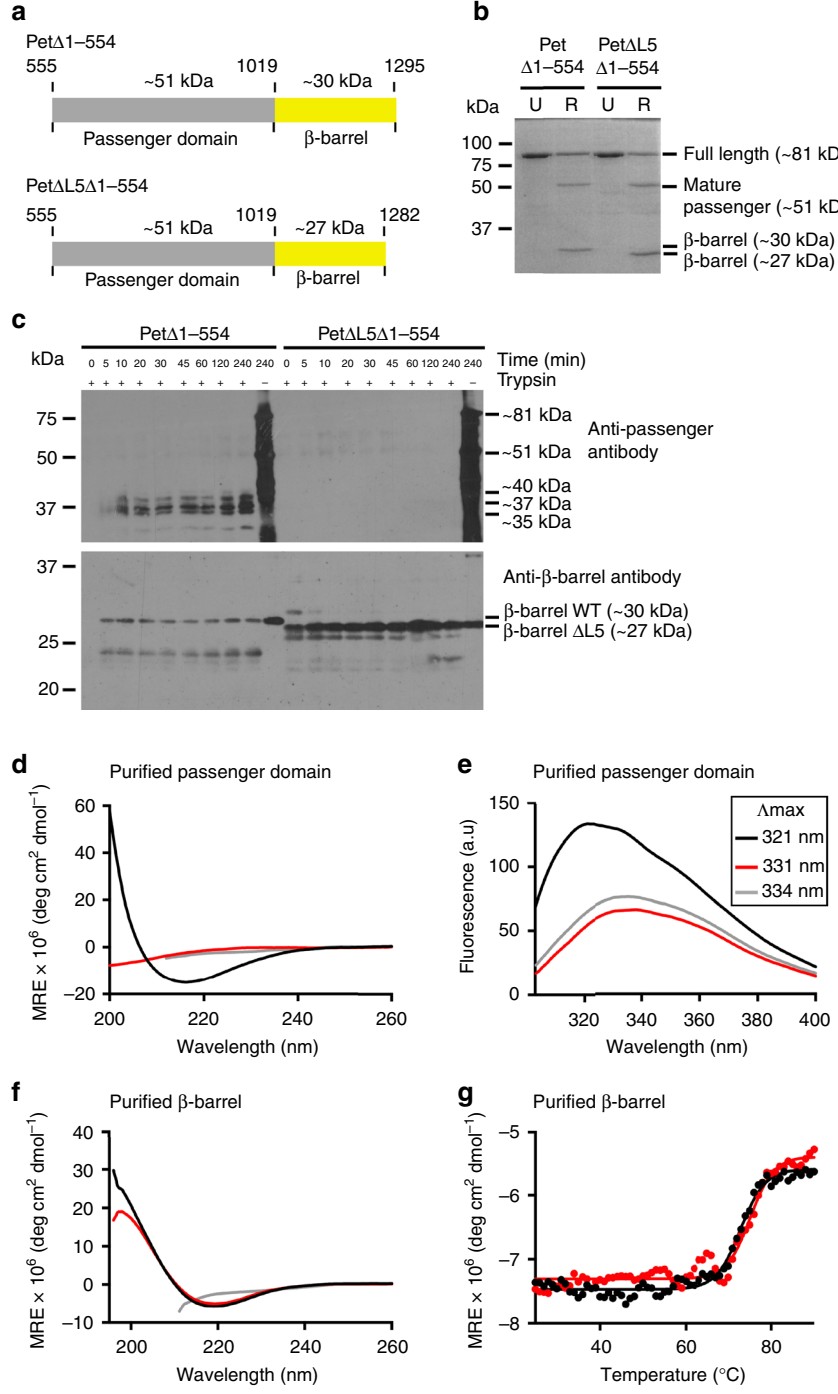

**Fig. 4** Truncation of L5 affects folding of the Pet passenger domain in vitro. **a** Schematic of the Pet$^{\Delta 1\text{-}554}$ and PetΔL5$^{\Delta 1\text{-}554}$ proteins. The 464-residue passenger incorporates residues 555 to 1018 of Pet and includes the α-helical segment, which connects this domain to the wild-type (277 residues; 1019 to 1295) or L5-truncated (264 residues; 1019 to 1282) β-barrel domain. Cleavage within the α-helical segment occurs between residues $N_{1018}$ and $N_{1019}$, with this autocatalytic cleavage converting the ~ 81 kDa unfolded, full-length protein into ~ 51 kDa and ~ 30/27 kDa proteins corresponding to the cleaved passenger and β-barrel domains, respectively. **b** Cells expressing Pet$^{\Delta 1\text{-}554}$ and PetΔL5$^{\Delta 1\text{-}554}$ were collected after induction with IPTG. These proteins were recovered from inclusion bodies, solubilized in urea, and then refolded by rapid dilution into LDAO detergent micelles. The refolding reaction was monitored by SDS-PAGE and Coomassie blue staining of the gels. Processing of the truncated passenger domain resulted in the autocatalytic conversion of the ~ 81 kDa full-length protein to ~ 51 kDa and ~ 30/27 kDa proteins, and served as an indicator of β-barrel folding. U, unfolded protein; R, refolded protein. **c** Folding of Pet$^{\Delta 1\text{-}554}$ (left) and PetΔL5$^{\Delta 1\text{-}554}$ (right) monitored by the accumulation of trypsin-resistant fragments. In this assay, folding was initiated and aliquots of the reaction mixture, at increasing incubation times, were treated with trypsin and then mixed with SDS to stop folding. Samples were analyzed by SDS-PAGE and immunoblotting using anti-passenger domain (top panel) and anti-β-barrel domain (bottom panel) antibodies. Far-UV CD (**d**) and tryptophan fluorescence (**e**) spectra of unfolded protein (grey), and in vitro folded, cleaved, and purified Pet$^{\Delta 1\text{-}554}$ (black) and PetΔL5$^{\Delta 1\text{-}554}$ (red) passenger domains. **f** Far-UV CD spectra of unfolded protein (gray), and in vitro folded, cleaved and purified Pet$^{\Delta 1\text{-}554}$ (black) and PetΔL5$^{\Delta 1\text{-}554}$ (red) β-barrel domains. **g** Thermal denaturation of in vitro folded, cleaved and purified Pet$^{\Delta 1\text{-}554}$ (black) and PetΔL5$^{\Delta 1\text{-}554}$ (red) β-barrel domains monitored in 1% SDS at 218 nm. MRE; mean residue ellipticity

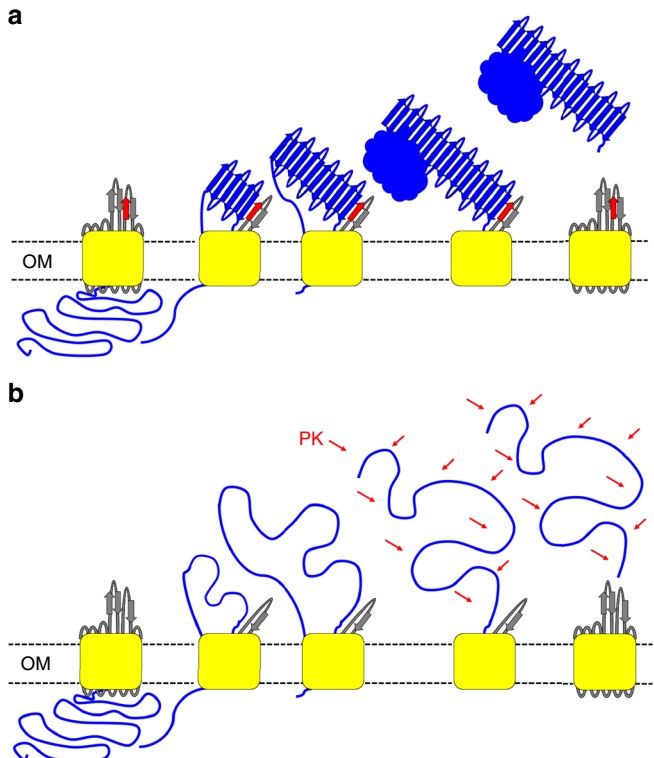

**Fig. 5** Proposed mechanism for passenger domain folding. **a** Schematic for folding shows that the first β-strand in L5 (red) interacts with the stable core in an antiparallel fashion during translocation, thereby polymerizing processive in-register stacking of the β-rungs and formation of the β-helix. Once the extreme N-terminus of the passenger domain has traversed the β-barrel pore, an autocatalytic cleavage reaction within an intra-barrel α-helical segment liberates the ~ 106 kDa mature form of the passenger domain, leaving the ~ 30 kDa β-barrel domain embedded in the outer membrane. **b** Schematic shows that in the absence of the L5 β-hairpin, the potential for β-strand augmentation is eliminated and thus interactions between the passenger domain and the first β-strand in L5 are prevented, thereby perturbing the formation of a continuous β-helix. Nevertheless, once the extreme N-terminus of the passenger domain has traversed the β-barrel pore, autocatalytic cleavage liberates the ~ 106 kDa mature form of the passenger domain, leaving the ~ 30 kDa β-barrel domain embedded in the outer membrane. As the passenger domain adopts an unfolded conformation, it is susceptible to degradation by host and exogenous proteases

C-terminal end of SurA, but not its specific amino acid sequence, are important for SurA function[52], led to the proposition that the chaperone function of SurA also necessitates β-strand augmentation with its substrates[50].

Although we have not yet identified the region of the passenger that transiently interacts with L5, we hypothesize that this region lies within the stable core given that these C-terminal β-helical rungs have been shown to appear first at the cell surface[3,19,23,24]. In support of this hypothesis are studies showing that mutation or deletion of the C-terminal β-helical rungs inhibits their folding[19] or the folding of the passenger N-terminus[37], respectively. We propose that the first β-strand in L5 hydrogen bonds with the C-terminal portion of the unfolded passenger domain in an anti-parallel manner, which initiates polymerization of the stable core into a stack of in-register β-helical rungs to which the remaining N-terminal β-helical backbone is built upon (Fig. 5a). Our observation that the β-strand propensity of the L5 β-hairpin, but not its specific amino acid sequence is important

for mediating folding of the Pet passenger domain suggests that the function of the L5 β-hairpin requires β-augmentation with the passenger domain, as opposed to recognition of a specific sequence. This conclusion is consistent with the greater amino acid sequence diversity found in SPATE passenger domains, which have amino acid identities ranging from 35% to 55%[42]. In the absence of the L5 β-hairpin, we find that the passenger domain mainly adopts an unfolded conformation and is therefore susceptible to degradation by host and exogenous proteases. We propose that when the potential for β-strand augmentation is eliminated, hydrogen bonding between the unfolded passenger and the first β-strand in L5 is prevented, and that it is the absence of these interactions at the very beginning of the nucleation process that perturbs the formation of a continuous β-helix (Fig. 5b).

Crystal structures of EspP show that L5 can adopt two different conformations[29,30]. The crystal structure of the EspP β-barrel domain post cleavage of the passenger domain (PDB code 2QOM) shows L5 in a 'closed' conformation, folded into the pore of the β-barrel where it hydrogen bonds with several β-strands. In contrast, the crystal structure of the EspP β-barrel domain with the most C-terminal portion of the passenger domain still attached (PDB code 3SLJ) shows L5 in an 'open' conformation, projecting into the extracellular space. Here, the uncleaved passenger domain traverses the length of the pore, thereby physically blocking L5 from making extensive contact with the β-barrel. Instead, the second β-strand in L4 interacts with the first β-strand in L5, forming a four-stranded β-sheet (Fig. 1b). Given the conformational flexibility of L5, we speculate that the L4 and L5 interaction is transient and that the β-hairpins separate during passenger domain translocation to allow L5 to interact with the passenger domain. In this scenario, L4 functions to restrain L5 movement and ensures that L5 is correctly positioned to interact with the passenger during translocation, thereby indirectly promoting folding of this domain. However, translocation occurs non-optimally through the L4-truncated β-barrel domain, suggesting a role for L4 in streamlining efficient passenger translocation. Hence, in an alternative scenario, L4 feeds the passenger domain to L5. Certainly, L4 forms part of a hydrophobic cavity in the BrkA β-barrel domain (which is also present in EspP) that is required to efficiently incorporate the passenger domain in a hairpin conformation, thereby promoting fast and efficient translocation through the β-barrel lumen[35].

L5-assisted folding of the passenger domain can explain the molecular basis for folding defective phenotypes observed in previous studies that investigated the in vitro folding of passengers in isolation from their cognate β-barrels. Studies on pertactin, a classical autotransporter from *B. pertussis*, demonstrated that although folding of the passenger N terminus requires the presence of the passenger C terminus[53], a folded C-terminus is not sufficient to promote processive folding of the N terminus[3,54]. It was further shown that refolding of the pertactin passenger into a β-helix in isolation is a slow ( > 10 h), concerted and multistep process involving the entire protein, rather than vectorial[3,54]. By contrast, autotransporter assembly in vivo occurs rapidly (~ 10 min)[18] and is in line with the typical doubling time of *E. coli* (~ 20 min). We conclude that the mechanistic differences between passenger folding with and without the β-barrel have transpired due to the inability to recapitulate in vitro the effects of the autotransporter β-barrel on passenger domain folding in the bacterial cell.

In addition to actively augmenting folding, as proposed previously[25], it is also feasible that the β-barrel accelerates passenger folding through promotion of vectorial folding during translocation. This would allow the passenger C terminus to explore its conformational space before the N terminus has been

translocated, thereby decreasing the probability of populating non-native structures. Such non-native structures likely cause slow in vitro refolding kinetics because they require extensive reorganization before the native conformation can be reached[54]. Vectorial folding would also ensure that the correct part of the passenger is able to interact with the first β-strand in L5 at the very beginning of the translocation process.

In summary, we have discovered that the SPATE β-barrel domain is a folding vector that can act independently of the cellular machinery to promote folding of the passenger domain on a biologically relevant timescale and we provide evidence to show that the efficiency of this process is dependent on the L5 β-hairpin. The significance of our work is evident in the inability of the passenger domain to fold into its biologically active form in the absence of its interaction with L5, a phenotype that could be exploited for therapeutic purposes.

## Methods

**Reagents and bacterial strains**. Bacteria were grown at 37 °C in Luria–Bertani broth and where necessary, the growth medium was supplemented with 30 µg mL$^{-1}$ kanamycin and/or 100 µg mL$^{-1}$ ampicillin, 0.2% (vol/vol) D-glucose, 0.02% (vol/vol) L-arabinose, or 0.5 mM isopropylthio-β-galactoside. Details of the *E. coli* strains (TOP10, BW25113, and BL21 [DE3]) and plasmids used in this study are presented in Supplementary Table 1, and primers used for plasmid construction are presented in Supplementary Table 2.

**Plasmid construction**. pBADPet has been described previously[34]. pBADEspP, pBADEspPΔL5, and all pBADPet derivatives except PetL5β1/P were synthesized de novo by GenScript. To construct PetL5β1/P, megaprimer PCR was performed as described previously[34,55]. Briefly, the round 1 PCR was performed using 500 ng of template DNA (pBADPet) with 1 µg mL$^{-1}$ each of primers MPL5β1Pro and HindIIIPetRv. The round 2 PCR was performed with 4 µg of megaprimer and 1 µg of primer SalIPetFw using 500 ng of template DNA (pBADPet). The resulting amplicon and target vector (pBADPet) were then digested with HindIII and SalI and ligated. To construct pETPet$^{Δ1-554}$ and pETPetΔL5$^{Δ1-554}$ the Pet β-barrel domain and the last 464 residues of the Pet passenger domain were amplified using 500 ng of template DNA (pBADPet and pBADPetΔL5) with 1 µg mL$^{-1}$ each of primers NdeIPet464Fw and XhoIPetRv. The subsequent amplicons and target vector (pET-22b + ) were then digested with NdeI and XhoI, and ligated for an in frame C-terminal hexahistidine-tag fusion. All DNA modifications were confirmed by sequencing.

**In vivo protein expression assays**. The pBAD expression system[56] was used to express protein as previously described[34]. In all cases, bacterial cultures were grown to an OD$_{600}$ of ~ 0.6 in media containing 0.2% glucose to repress protein synthesis prior to harvesting (2500 × *g*, 5 min, 4 °C). For pulse-chase expression of the pBADPet derivatives, the bacterial cells were resuspended in media containing 0.02% arabinose for five minutes (pulse) to induce Pet expression, collected, and then resuspended in media containing 0.2% glucose to "chase" the protein produced during the "pulse" phase of the experiment through the biogenesis pathway. Throughout the pulse-chase assay, three 1 mL aliquots were removed from each culture at each time point. The first aliquot was added to a tube containing trichloroacetic acid (TCA; final concentration, 10% [wt/vol]), and placed on ice. The second aliquot was added to a tube containing proteinase K (final concentration, 200 µg mL$^{-1}$), and placed on ice for 20 min to digest Pet exposed on the cell surface. The protease reaction was stopped by the addition of 2 mM phenylmethanesulfonyl fluoride (PMSF) before TCA precipitation. All TCA precipitated samples were washed with acetone, dried, and resuspended in 100 µL of SDS-PAGE loading buffer per 1 OD$_{600}$ units of cells (third aliquot). Where necessary, a fourth aliquot was treated with 200 µg mL$^{-1}$ polymyxin B for 10 min on ice prior to proteinase K treatment and TCA precipitation. Ten microliters of the normalized samples were separated by SDS-PAGE on 3–14% gradient gels and subjected to immunoblotting using anti-Pet passenger domain antibodies at a 1:5,000 dilution. These antibodies have been described and validated previously[28,34].

Bacterial cultures harbouring pBADEspP and pBADEspPΔL5 were resuspended in arabinose for 30 min to induce EspP and EspPΔL5 expression. The OD$_{600}$ of cultures were normalized (to permit comparison of secreted protein levels) and pelleted before filtering of the supernatant fractions through 0.22 µm pore-size filters. A 10% (w/v) final concentration of TCA was then used to precipitate the secreted proteins as described above. Samples were separated by 10% SDS-PAGE and detected by staining with Coomassie Brilliant Blue R250. Whole cell lysates were separated by SDS-PAGE on 3–14% gradient gels and subjected to immunoblotting using anti-Pet β-barrel domain antibodies at a 1:5,000 dilution as described above. The Pet and EspP β-barrel domains are so similar that these

antibodies cross-react with the EspP β-barrel domain. These antibodies have been described and validated previously[34,57].

**Heat modifiability assay**. After pulse-chase protein expression, cells were harvested and prepared for heat modifiability experiments as described previously[34]. Briefly, 40 µg of total membranes were mixed with SDS-PAGE loading buffer and then heated at temperatures between 25 °C and 90 °C. Samples were separated by SDS-PAGE on 3–14% gradient gels and subjected to immunoblotting using anti-Pet β-barrel domain antibodies as described above.

**Protein refolding and purification**. Protein expression and refolding were carried out as previously described[34]. Briefly, *E. coli* BL21 (DE3) cells transformed with pETPet$^{Δ1-554}$ and pETPetΔL5$^{Δ1-554}$ were harvested (4200 × *g*, 5 min, 4 °C) after a 4 h (pETPet$^{Δ1-554}$) or 18 h (pETPetΔL5$^{Δ1-554}$) induction with 0.5 mM IPTG. Inclusion bodies were isolated by centrifugation (10,000 × *g*, 15 min, 4 °C) and solubilized in 8 M urea. Pet$^{Δ1-554}$ and PetΔL5$^{Δ1-554}$ were then refolded from the urea extracts by rapid 10-fold dilution into tris-buffered saline containing 0.5 % (w/v) lauryldimethylamine-oxide (LDAO) detergent micelles at 35 °C. Affinity purification with Ni-NTA resin was used to separate the refolded passenger and β-barrel domains. Addition of Bio-Beads$^{TM}$ SM-2 Resin (BioRad) at 0.1 g mL$^{-1}$ to the unbound flow-through containing the cleaved passenger domain removed the LDAO detergent and residual β-barrel. Subsequent gel filtration chromatography in 50 mM Tris pH 8.0, 150 mM NaCl (Superdex s200-10/300, GE HealthCare) resulted in purification of the ~ 51 kDa passenger domain. Bound β-barrel was eluted from the Ni-NTA resin and further purified by gel filtration chromatography in 50 mM tris pH 8.0, 150 mM NaCl, 0.05% LDAO (wt/vol) (Superdex s200–10/300, GE HealthCare), which resulted in purification of the folded ~ 30/27 kDa forms.

**Biophysical analysis of protein structure**. Kinetics of trypsin cleavage profiles during Pet$^{Δ1-554}$ and PetΔL5$^{Δ1-554}$ refolding were initiated as described above and were followed by limited treatment with trypsin, which was performed as described previously[34]. Briefly, aliquots of the reaction mixture were removed at the time points indicated and treated with trypsin (20 µg mL$^{-1}$) for 20 min on ice before quenching digestion and folding with 2 mM PMSF and SDS loading buffer, respectively. The resulting trypsin fragments were separated by SDS-PAGE and subjected to immunoblotting using anti-passenger and anti-β-barrel domain antibodies as described above.

Far-UV CD spectra were collected from 190 to 260 nm using a Chirascan spectrometer. Samples containing 0.1 mg mL$^{-1}$ of purified protein were measured at room temperature with a 1 mm cuvette, 1.0 nm bandwidth, 1 s integration time, and 20 nm/min scanning speed. Three scans were averaged, and the spectra were corrected for buffer contribution. For thermal denaturation measurements, the denaturation of β-sheet structure was monitored in the presence of 1% SDS at 218 nm as a function of temperature from 25 °C to 95 °C at a temperature change of 1 °C min$^{-1}$. Thermal unfolding was irreversible. In both cases, the CD spectra were normalized to the mean residue molar ellipticity.

A Varian Cary Eclipse fluorimeter was used to collect emission spectra between 300–400 nm upon excitation of tryptophan at 295 nm. Samples containing 0.01 mg mL$^{-1}$ of purified protein were collected at room temperature with a 1 cm cuvette, 5 nm slit width, 0.1 s integration time, and an increment of 1.0 nm. Three scans were averaged, and the spectra were subtracted for buffer contribution.

**Data availability**. All data generated or analysed during this study are included in this published article and its Supplementary Information files. The raw image files of the cropped immunoblots and Coomassie-stained gels displayed in the Figures and Supplementary Figures can be found in Supplementary Fig. 9. Other data are available from the corresponding author upon reasonable request.

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

## Acknowledgements

We thank Alex Joule and Anthony Newman for technical assistance, Trevor Lithgow and Iain Hay for providing the *E. coli* BW25113ΔompT strain, John Carver for access to the Chirascan spectrometer, as well as Susan Howitt for critical comments on the manuscript. We gratefully acknowledge support of the Australian Research Council (ARC) for

research funding through the ARC Discovery Project grant DP160103294 (to D.L.L. and I.R.H.). M.A.S. is a National Health and Medical Research Council Senior Research Fellow (APP1106930), G.H.M.H. is supported by a Global Marie-Curie Fellowship (EU project 660083), and D.L.L. is an ARC Future Fellow (FT150100452).

## Author contributions

X.Y., M.D.J., G.H.M.H., I.R.H., and D.L.L. conceived and designed the experiments. X.Y., M.D.J., J.Z., and A.W.L. performed the experiments. X.Y., M.D.J., G.H.M.H., A.W.L., M.A.S., L.C.W., R.N.P., I.R.H., and D.L.L. analyzed the data. X.Y., M.D.J., and D.L.L. wrote manuscript. All authors have reviewed the manuscript.

## Additional information

**Competing interests:** All authors declare no competing interests.

