## [Peer Review File(PDF 653 kb) · Nature Communications]

Reviewers' comments:

Reviewer #1 (Remarks to the Author):

Autotransporters consist of a large N-terminal 'passenger' domain and a C-terminal beta-barrel domain that facilitates transport of the passenger across the bacterial outer membrane. Yuan and co-workers look into the function of some prominent extracellular loops present on autotransporter beta-barrels. Intriguingly, they find that loop L5, which forms a so-called beta-hairpin, is critical for autotransporter passengers to reach their native conformation upon translocation. The data translate into an exciting model in which interactions between the L5 beta-hairpin and the most C-terminal part of autotransporter passengers nucleates folding of the passengers into their typical beta-helical conformation via beta-strand augmentation.

This is very interesting work providing highly valuable insight into the biogenesis of bacterial autotransporters, which is still a largely enigmatic process. Overall, the experiments are well conceived, but additional control experiments are needed. Also, additional rationale with some experiments and better explanation of results at places would enhance the quality of the manuscript. One important issue that should be addressed concerns the potential role of loop L4 in the proposed mechanism, which seems somewhat neglected and incompletely analyzed.

Major comments:

- line 93,94: The crystal structure of EspP was used for topology modeling of the Pet beta-domain. Was the modeling done 'by hand' or aided by software? Please provide details about the software used and the underlying considerations. Why does the topology model not follow the cited EspP barrel crystal structure (pdb: 3slj), which shows that L4 and L5 are not independent loops but together form a 4-stranded beta-sheet? It seems more appropriate to align L4-L5 such that polar contacts between these loops that are visible in the pdb file are represented.
- Based on the Pet deletion mutant tested in Supp Fig. 1a/b the authors draw the conclusion that L4 is not critical for passenger folding, but only the tip of the loop has been removed, with half of the beta-hairpin still intact. In contrast, to study the role of L5, the complete loop beta-hairpin was deleted. What is the rationale behind deleting only part of L4? The remainder of L4 might still play a role in passenger folding, either directly or indirectly. An L4 mutant lacking at least the complete beta-hairpin should be tested.
- In the proposed beta-augmentation model, the first beta-strand of L5 interacts with the passenger to nucleate passenger folding. How do the authors reconcile this with the crystal structure of EspP that shows that the first beta-strand of L5 is engaged in beta sheet formation with L4?
- Lines 119-125; Fig. 1e, left panel: Authors state that FL Pet is converted into a 70 kDa protease resistant fragment upon PK treatment and therefore claim that the passenger already attains a native conformation while still attached to the beta-barrel. This is questionable. Perhaps the lowered amount of FL product is due to full degradation of the FL species by PK and all 70 kDa material visible on blot is derived from cleaved passenger that is already present in ample amounts at the concerning timepoints. More evidence is needed if the authors want to claim that the passenger of FL Pet product already attained a PK resistant conformation.
- Fig. 1e/f: -PK and +PK samples should derive from the same culture sample. How do the authors explain that FL Pet is visible already after 2 min in the +PK samples and only after 5 min in the -PK samples?
- Lines 124-128; Fig. 1e: Authors state that FL Pet Δ L5 is surface exposed and in unfolded conformation, judged by its sensitivity towards PK treatment. An immunoblot against a PK-sensitive periplasmic protein should be carried out on the same samples to show that the cells were intact during the PK procedure and that the FL Pet detected is really surface exposed and not intracellular off-pathway product. Same goes for samples Fig. 1f.

- Fig. 1e/f: Were the same amounts of cell material loaded and the same exposure times used for blots of corresponding Pet and Pet Δ L5 samples? As beta helix formation is thought to provide the driving force for passenger translocation, export of Pet Δ L5 could and, in turn, result in partial degradation of the protein in the periplasm. Stating loaded amounts and exposure times are important to judge this aspect.
- Fig. 1g: With regard to the above, were identical amounts of membrane material loaded and the same exposure times used for the blots?
- Fig. 1f. The experiment in ompT-minus background needs to be supplemented with pulse-chase/PK analysis in the isogenic wild-type strain: BW25113. In contrast to TOP10, BW25113 derivatives carry a deletion of the araBAD operon and, therefore, responded differently to arabinose, which was used for induction of Pet expression. As a consequence, there may be differences between TOP10 and BW25113-derived strains in terms of expression levels/kinetics of Pet Δ L5 and the handling of potential off-pathway products.
- Lines 196-210; Fig. 2d: Authors state that PetL5 β 1/P and PetL5OmpF are deficient in passenger folding, since no protease resistant fragment emerges upon PK treatment. However, the fact that the FL species of these constructs are largely resistant against PK – particularly for PetL5 β 1/P – seems to indicate that these constructs are largely translocation incompetent, rather than affected in passenger folding per se. Additional explanation or evidence is needed to substantiate the authors' interpretations.
- Lines 207-210: Length of the L5 beta-hairpin as a critical parameter is based on the observation that a 4 aa hairpin of OmpF does not mediate passenger folding as opposed to the longer hairpins of Pet and FadL. Rather than using a heterologous OmpF hairpin (that seems to interfere with upstream processes; see comment above), constructs carrying shortened versions of the endogenous Pet L5 hairpin should be used to make this point.
- Lines 258-263: "In contrast, the EstA passenger domain has a globular fold where a central four-stranded parallel β -sheet interacts with the L5 β -hairpin (Supplementary Fig. 4a)..... structure-based analysis makes way for a conserved role for L5 in the folding of SPATE and non-SPATE passenger domains, regardless of whether the passenger fold is β -helical or globular with mixed α/β - content." The interactions between L5 and the beta-sheet are not apparent in the EstA pdb file (3kvn). In any case, a conserved role for L5 in passengers outside of the SPATE subfamily is likely but remains speculative unless actual experiments have been carried out with mutants lacking L5. This could be done for EstA and Ag43 which have been well described in literature and for which reagents such as antibodies are available.
- Data on the autotransporter EspP lacking L5 have been published (Barnard et al., Nat Struct Mol Biol. 2007) and should be discussed in the context of the current work.

Minor comments:

- line 71: remove space between (n- n)
- line 117: "full-length Pet": please make clear that this refers to the species without signal sequence in this case
- Fig. 1e, right panel, -PK: please align '0, 1, 2, 5, 10 15 min' annotation better with lanes. Confusing.
- Fig. 1g looks a bit sloppy, which disturbs interpretation. Much clearer pictures of the same assay were produced in previous publications by the authors.
- Supp Fig. 5a: SDS-PAGE lanes Pet Δ L5 Δ 1-554 looks strange: half of the right hand lane seems to be missing
- Lines 445-450: concentration or % glucose and arabinose used should be specified.

Reviewer #2 (Remarks to the Author):

Autotransporters are central virulence factors in the pathogenesis of many Gram-negative bacterial

species. These proteins are composed of a translocator domain in the bacterial outer membrane and a passenger domain that is transported to the extracellular milieu. Folding of the passenger domain is a key event in secretion and activity. Folding occurs only once the passenger domain is exposed on the extracellular surface, but it remains unknown what triggers this reaction. In this work, Leyton and colleague discover that the region corresponding to the extracellular loop 5 of the translocator domain is essential for passenger domain folding. By using *in vivo* and *in vitro* experimental approaches they conclude that the β -hairpin conformation of loop 5 functions in nucleating the β -helical folding of the passenger domain.

This is a well written manuscript that will have a broad impact in the field of protein secretion. The authors show clear evidence that loop 5 is necessary for correct passenger domain folding, which is a novel function of the autotransporter translocator domain. The conclusion that its β -hairpin structure nucleates folding of the β -helical passenger domain remains an attractive hypothesis and not a proven fact, as it might appear from reading the abstract (see second last sentence) and other parts of the text. Even in the absence of such proof, this remains a relevant piece of work and it should be considered for publication once the authors address the following points.

1) It appears odd that the authors compare the biogenesis of Pet in the BW25133 Δ OmpT and the Top10 strain. A direct comparison of pulse-chase Pet expression in BW25133 and BW25133 Δ ompT should be shown.

2) Differently from what stated by the authors (page 7, Figure S1), deletion of loop4 delays of at least 5 min passenger domain cleavage and the appearance of its 70 kDa PK resistant fragment. Does loop 4 influence the activity of loop 5 in promoting correct passenger domain folding? In the absence of loop 5, may loop 4 be incorrectly positioned and interfere with the folding of the passenger domain. The authors should test the biogenesis of a Pet construct that lacks both loop 4 and loop 5.

3) PetL5 β 1/G, PetL5OmpF, PetL β 1/P: Processing of the passenger domain cannot be detected. To which extent is the passenger domain secreted? Would the passenger domain accumulate in the BW25133 Δ ompT strain? PK accessibility could be due to an improperly folded translocator domain?

4) What would be the effect of replacing Pet loop5 with an α -helical hairpin similar to that of FadL loop 3? This would be an interesting experiment, which might (or might not) support the folding nucleation hypothesis

5) Deletion of loop 5 does not affect the stability of the translocator domain. Why is the amount of the EspP β -barrel lacking loop5 lower than the amount of the wild type β -barrel domain in figure 3b? What is the overall level of expression/stability of the two constructs used in this figure?

6) The authors use an elegant *in vitro* approach to prove the critical role of loop 5 in the folding of the passenger domain. Certainly this experiment is the strongest evidence in favor of the folding nucleation hypothesis. To further reinforce this point, and given that this *in vitro* approach is already set-up in their labs, the authors should investigate whether a short polypeptide corresponding to the loop5 β hairpin has any *in trans* activity on the folding of the purified passenger domain.

7) Would it be possible that loop 5 influences the correct positioning of the C-terminal region of the passenger domain, which has to be at a certain distance (or orientation) from the β -barrel domain in order to explore different possible conformations and acquire correct folding? Proving that the β -hairpin of loop 5 nucleates folding of the passenger domain will be extremely challenging. This is currently the main hypothesis discussed by the authors. Other alternatives concerning the molecular

role of loop 5 need to be considered.

Reviewer #3 (Remarks to the Author):

This is a very clearly written manuscript on passenger domain folding in autotransporters, and specifically on the influence of an outer membrane surface loop (L5) on this complex protein folding and transport process. I have some suggestions on how to improve the manuscript, and I do think that additional controls are necessary to be able to generalize claims about this - clearly important - loop region.

- to put the work in perspective, I would suggest to mention (maybe in the abstract and certainly in the introduction) that there are different classes of autotransporters, and that the work presented here is only done on one of them - the "classical" autotransporters, of type Va secretion systems. Other autotransporter have different barrel features, including oligomerization, or the barrels being at the N-terminus. The current phrasing does not clarify this, but rather makes it sound as if the features described in this manuscript are universal (e.g. the barrel being at the C-terminus...). All of this is probably included in the citations used by the authors, but should be mentioned explicitly for the non-expert reader. This is also relevant for the discussion part where the authors generalize the importance of the effects found here, but make it sound as if this is true for all autotransporters (instead of one class)

Regarding the experimental work, I have a few comments that might need addressing:

- several of the blots are of poor quality. Specifically figure 1g is very smeary, and figure 4c is totally overloaded in 2 of the lanes.

- I am missing an (obvious?) control in the experiments: a control for periplasmic protection in the PK digestion. The experiments with the ompT deletion mutant suggest that PetdeltaL5 is surface exposed, but to rule out periplasmic degradation by PK (due to loss of OM integrity in the mutant), the authors should show a blot of a periplasmic protein (maybe SurA or similar, or something like MBP? The latter can be expressed easily in E.coli and exists in variants of different stability) that is unaffected by PK in both the WT and the mutant.

- to make a general claim that L5 is "the nucleation site for passenger folding" in other types of ATs as well (and again: only the type Va ones?), maybe this should be tested directly, e.g. by replacing the loop in their system with one from a non-SPATE autotransporter (similar to the FadL replacement shown here)?

- Why is there no signal for the passenger in the trypsinated refolding experiments at time point 0? Should there not be a clear signal? This needs some discussion I think?

Point-by-point rebuttal for NCOMMS-17-19249-T

We thank the Reviewers for their positive appraisals of the paper. We found their comments to be constructive and have addressed the issues raised with new experiments and with revision to the text (shown in red font), as detailed point-by-point below:

Reviewer #1 (Remarks to the Author):

Autotransporters consist of a large N-terminal ‘passenger’ domain and a C-terminal beta-barrel domain that facilitates transport of the passenger across the bacterial outer membrane. Yuan and co-workers look into the function of some prominent extracellular loops present on autotransporter beta-barrels. Intriguingly, they find that loop L5, which forms a so-called beta-hairpin, is critical for autotransporter passengers to reach their native conformation upon translocation. The data translate into an exciting model in which interactions between the L5 beta-hairpin and the most C-terminal part of autotransporter passengers nucleates folding of the passengers into their typical beta-helical conformation via beta-strand augmentation.

This is very interesting work providing highly valuable insight into the biogenesis of bacterial autotransporters, which is still a largely enigmatic process. Overall, the experiments are well conceived, but additional control experiments are needed.

We thank the Reviewer for picking up this important point. We have now included all of the additional control experiments requested as detailed below.

Also, additional rationale with some experiments and better explanation of results at places would enhance the quality of the manuscript.

We hope that we have addressed these concerns of the Reviewer throughout the text. While it is difficult to pinpoint where the Reviewer would like additional explanations/rationale, we have modified the text where this and other Reviewers had specific questions. We note that, overall, Reviewers 2 and 3 stated that the text was “well” and “clearly” written and hope that our modifications also please Reviewer 1.

One important issue that should be addressed concerns the potential role of loop L4 in the proposed mechanism, which seems somewhat neglected and incompletely analyzed.

We now discuss the potential role of L4 in more detail and touch upon its role with respect to passenger translocation and folding. Upon Reviewer requests, we have also added further experiments with respect to L4. These are detailed below (Reviewer 1, point 2 and Reviewer 2, point 2).

Major comments:

1- line 93,94: The crystal structure of EspP was used for topology modeling of the Pet beta-domain. Was the modeling done ‘by hand’ or aided by software? Please provide details about the software used and the underlying considerations. Why does the topology model not follow the cited EspP barrel crystal structure (pdb: 3slj), which shows that L4 and L5 are not independent loops but together form a 4-stranded beta-sheet? It seems more appropriate to align L4-L5 such that polar contacts between these loops that are visible in the pdb file are represented.

Topology modelling of the Pet β -barrel domain was done by hand based on the boundaries defined by the crystal structure of the EspP β -barrel domain (PDB code 3SLJ). The fact that the β -barrel domains of both proteins are 277 amino acid residues in length and that these β -barrels share ~91% sequence identity rendered the process of manual topology modelling simple and straightforward.

Our topology model does not show that L4 and L5 form a 4-stranded β -sheet as shown in the crystal structure of the EspP β -barrel domain with PDB code 3SLJ because these loops are mobile and therefore not always in this conformation. For example, the original crystal structure of the EspP β -barrel domain (PDB 2QOM) shows L5 folded into the pore of the β -barrel. Since this crystal structure shows the β -barrel post-cleavage and release of the passenger, the authors (Barnard et al., 2007 NSMB) proposed that L5 in this ‘closed’ conformation restricts solvent access into the bacterium from the extracellular side. In this crystal structure, L3 and L4 remain partially unresolved. In contrast, the EspP β -barrel domain cited in our work (PDB code 3SLJ) shows a pre-cleavage structure (i.e. the most extreme C-terminal portion of the passenger domain is still attached) where L5 is in the ‘open’ conformation, projecting into the extracellular space. The authors (Barnard et al., 2012 J Mol Biol) show that the uncleaved passenger domain, which traverses the

length of the pore, also projecting into the extracellular space, physically blocks L5 from making extensive contact with the β -barrel. Since passenger domain translocation precedes cleavage (Skillman et al., 2005 Mol Microbiol) and folding of the passenger domain is proposed to be vectorial (Braselman & Clark, 2012 J Phys Chem Lett), we used the EspP β -barrel domain with PDB code 3SLJ in our work because it better represents the conformation of L5 during translocation. Nevertheless, given that L3, L4 and L5 are mobile and the equilibrium solution conformations are unknown, we prefer not to align L4 and L5 such that the polar contacts between these two loops are visible.

2- Based on the Pet deletion mutant tested in Supp Fig. 1a/b the authors draw the conclusion that L4 is not critical for passenger folding, but only the tip of the loop has been removed, with half of the beta-hairpin still intact. In contrast, to study the role of L5, the complete loop beta-hairpin was deleted. What is the rationale behind deleting only part of L4? The remainder of L4 might still play a role in passenger folding, either directly or indirectly. An L4 mutant lacking at least the complete beta-hairpin should be tested. All three Pet loop deletion mutants were designed by co-author Ian Henderson in 2006 based on the crystal structure of the NalP β -barrel domain (1UYN), which was published in 2004 (Oomen et al., 2004 EMBO). This occurred prior to publication of the NSMB paper by Barnard and co-authors reporting the crystal structure of the EspP β -barrel domain in 2007. Since the β -barrel domains of Pet and NalP only share ~18% sequence identity, the predicted boundaries of some of the Pet loops were not precise. Nevertheless, we agree with the Reviewer's point and have added to Supplementary Fig. 2, pulse-chase assays of a new L4 truncation mutant in *E. coli* TOP10 showing that complete removal of the L4 β -hairpin results in a substantial reduction of passenger domain translocation as shown previously for autotransporter BrkA (see Zhai et al., 2011 Biochem J). Pulse-chase expression of the full L4 truncation mutant in *E. coli* BW25113 $\Delta ompT$ showed an increase in cleavage efficiency, suggesting that part of the passenger population is unfolded. Since our original data using a L4 mutant with a partially deleted β -hairpin shows a minor translocation defect, but no obvious folding defect, these data suggest that L4 mainly acts during translocation and that if L4 has a role in passenger folding, its role during folding is non-essential. We have revised the Discussion with these new data in mind.

3- In the proposed beta-augmentation model, the first beta-strand of L5 interacts with the passenger to nucleate passenger folding. How do the authors reconcile this with the crystal structure of EspP that shows that the first beta-strand of L5 is engaged in beta sheet formation with L4? As explained above, L4 and L5 are mobile and not always in close proximity. Indeed, PDB 2QOM suggests that polar contacts between L4 and L5 are not restored post passenger translocation, cleavage and release, where L5 instead closes off the β -barrel. While speculative, we propose that the polar contacts between L4 and L5, as depicted in PDB 3SLJ, are broken during passenger domain translocation to allow L5 to transiently interact with the passenger domain. We have revised the Discussion with this in mind.

4- Lines 119-125; Fig. 1e, left panel: Authors state that FL Pet is converted into a 70 kDa protease resistant fragment upon PK treatment and therefore claim that the passenger already attains a native conformation while still attached to the beta-barrel. This is questionable. Perhaps the lowered amount of FL product is due to full degradation of the FL species by PK and all 70 kDa material visible on blot is derived from cleaved passenger that is already present in ample amounts at the concerning timepoints. More evidence is needed if the authors want to claim that the passenger of FL Pet product already attained a PK resistant conformation. We agree with the Reviewer's point and have modified the text on p. 6 to make it clear that it is likely the digestion of mature Pet passenger domain molecules with proteinase K that have resulted in the generation of the ~70 kDa protease-resistant fragment.

5- Fig. 1e/f: -PK and +PK samples should derive from the same culture sample. How do the authors explain that FL Pet is visible already after 2 min in the +PK samples and only after 5 min in the -PK samples? As described in the Methods section, +PK and -PK samples are indeed from the same culture samples. However, -PK samples are removed from each culture at each time point and immediately treated with trichloroacetic acid (TCA), which denatures proteins and fixes bacteria, thereby halting Pet biogenesis. In contrast, +PK samples are placed on ice for 20 min to digest Pet exposed on the cell surface. Importantly, unlike TCA, proteinase K does not denature proteins or fix bacteria, but does digest Pet exposed on the cell surface when used at a final concentration of 200 $\mu\text{g}/\text{mL}$. This means that samples treated with proteinase K

have an additional 20 min of Pet biogenesis (although this is substantially slowed down by the samples being incubated on ice), which explains why full-length Pet is visible after 2 min in the +PK samples, but only present after 5 min in the –PK samples.

6- Lines 124-128; Fig. 1e: Authors state that FL Pet Δ L5 is surface exposed and in unfolded conformation, judged by its sensitivity towards PK treatment. An immunoblot against a PK-sensitive periplasmic protein should be carried out on the same samples to show that the cells were intact during the PK procedure and that the FL Pet detected is really surface exposed and not intracellular off-pathway product. Same goes for samples Fig. 1f.

We have shown previously that in *E. coli* TOP10, the amount of proteinase K in use is sufficient to cleave chaperone SurA and the lipoprotein BamD, both of which reside in the periplasm, but only in the presence of polymyxin B, an antibiotic that selectively permeabilizes the outer membrane (see Leyton et al., 2014 Nat Commun). Nevertheless, we have added as Supplementary Fig. 1a and 1b, pulse-chase assays showing that in *E. coli* TOP10 and *E. coli* BW25113 Δ ompT, the intact SurA protein (48 kDa) is observed in –PK and +PK samples. These data show that the amount of proteinase K in use does not permeabilize the *E. coli* outer membrane, which remains intact during proteinase K treatment. Additional treatment of *E. coli* TOP10 with polymyxin B and proteinase K degraded SurA almost to completion. In contrast, intact SurA was observed in samples treated with proteinase K in the presence of polymyxin B in *E. coli* BW25113 Δ ompT, indicating that 200 mg/mL polymyxin B is insufficient to permeabilize this strain's outer membrane. As a consequence, only *E. coli* TOP10 was used to perform pulse-chase assays in the presence of polymyxin B and proteinase K in this study. Importantly, pulse-chase assays of *E. coli* BW25113 Δ ompT in the presence of polymyxin B and proteinase K were not necessary for the interpretation of our results, which is why titration of polymyxin B to identify the amount required to permeabilize this strain's outer membrane was not pursued. In addition, we have modified the text on p. 5 to make it clear that the *E. coli* cells used in the experiments shown in Fig. 1e and 1f were indeed intact during treatment with proteinase K, unless first treated with polymyxin B, and have included the methodology used for Polymyxin B treatment in the Methods section (see p. 22).

7- Fig. 1e/f: Were the same amounts of cell material loaded and the same exposure times used for blots of corresponding Pet and Pet Δ L5 samples? As beta helix formation is thought to provide the driving force for passenger translocation, export of Pet Δ L5 could and, in turn, result in partial degradation of the protein in the periplasm. Stating loaded amounts and exposure times are important to judge this aspect.

The pulse-chase assays performed in this study were carried out exactly as described previously (see Leyton et al., 2014 Nat Commun). We have modified the text in the Methods (see p. 23) to specify that equivalent amount (10 μ L) of density-normalised bacterial samples were separated by SDS-PAGE on 3-14% gradient gels and subjected to immunoblotting. All directly comparable blots were developed in the same cassette, at the same time, and for the same amount of time (e.g. Fig. 1e, left and right panels, Fig. 1g, top and bottom panels, etc.). However, due to the technically demanding and time sensitive nature of our pulse-chase assay, a maximum of three assays were ever done at the same time, which included one or two Pet mutants always in conjunction with wild type Pet. The blots shown in this current study were exposed for different lengths of time because different batches of Protein-A-Sepharose-purified antibodies (with different efficacies) were used over the duration of the study. Where the immunoblotting banding patterns for wild type Pet looked indistinguishable between assays, immunoblots for mutant Pet variants were grouped with one immunoblot of wild type Pet for presentation purposes (e.g. Fig. 2a,b). We will provide raw uncropped immunoblots for all pulse-chase assays should our manuscript be accepted.

8- Fig. 1g: With regard to the above, were identical amounts of membrane material loaded and the same exposure times used for the blots?

Yes, identical amounts of total membrane proteins were loaded and the same exposure times were used for the blots. However, we agree with the Reviewer's Minor comment no. 4 in that this figure looks a bit sloppy compared to similar figures that we have published recently (see Leyton et al., 2014 Nat Commun). We have therefore repeated the assay and have replaced the original Fig. 1g with a new heat-modifiability assay that confirms our original data, i.e. that truncation of L5 does not inhibit folding or insertion of the β -barrel into bacterial outer membranes, or autocatalytic release of the passenger domain into the culture supernatant.

9- Fig. 1f. The experiment in ompT-minus background needs to be supplemented with pulse-chase/PK analysis in the isogenic wild-type strain: BW25113. In contrast to TOP10, BW25113 derivatives carry a deletion of the araBAD operon and, therefore, responded differently to arabinose, which was used for induction of Pet expression. As a consequence, there may be differences between TOP10 and BW25113-derived strains in terms of expression levels/kinetics of Pet Δ L5 and the handling of potential off-pathway products.

We thank the Reviewer for picking up on this point. We have added as Supplementary Fig. 3, a pulse-chase assay of Pet and Pet Δ L5 in wild type *E. coli* BW25113, which shows that there are no differences between *E. coli* TOP10 and *E. coli* BW25113 in terms of Pet Δ L5 expression levels, kinetics, and in the handling of off-pathway products. In other words, the pulse-chase expression of Pet and Pet Δ L5 in *E. coli* TOP10 and *E. coli* BW25113 occur in a similar fashion.

10- Lines 196-210; Fig. 2d: Authors state that PetL5 β 1/P and PetL5OmpF are deficient in passenger folding, since no protease resistant fragment emerges upon PK treatment. However, the fact that the FL species of these constructs are largely resistant against PK – particularly for PetL5 β 1/P – seems to indicate that these constructs are largely translocation incompetent, rather than affected in passenger folding per se. Additional explanation or evidence is needed to substantiate the authors' interpretations.

We thank the Reviewer for picking up on this important point. We have added as Supplementary Fig. 4, pulse-chase assays of Pet^{L5 β 1/G}, Pet^{L5 β 1/P} and Pet^{L5OmpF} in *E. coli* BW25113 Δ ompT, which rule out a translocation defect as an explanation for the absence of the ~70 kDa proteinase-resistant fragment, thereby strongly supporting the hypothesis that β -strand structure in L5 is required for passenger folding.

11- Lines 207-210: Length of the L5 beta-hairpin as a critical parameter is based on the observation that a 4 aa hairpin of OmpF does not mediate passenger folding as opposed to the longer hairpins of Pet and FadL. Rather than using a heterologous OmpF hairpin (that seems to interfere with upstream processes; see comment above), constructs carrying shortened versions of the endogenous Pet L5 hairpin should be used to make this point.

We thank the Reviewer for this suggestion. We have added as Supplementary Fig. 6, pulse-chase assays of a series of L5 truncation mutants, which show that the endogenous β -hairpin must be of a certain length (comprising of no less than five residues) to scaffold folding of the Pet passenger domain.

12- Lines 258-263: "In contrast, the EstA passenger domain has a globular fold where a central four-stranded parallel β -sheet interacts with the L5 β -hairpin (Supplementary Fig. 4a)..... structure-based analysis makes way for a conserved role for L5 in the folding of SPATE and non-SPATE passenger domains, regardless of whether the passenger fold is β -helical or globular with mixed α/β - content." The interactions between L5 and the beta-sheet are not apparent in the EstA pdb file (3kvn). In any case, a conserved role for L5 in passengers outside of the SPATE subfamily is likely but remains speculative unless actual experiments have been carried out with mutants lacking L5. This could be done for EstA and Ag43 which have been well described in literature and for which reagents such as antibodies are available.

We appreciate the Reviewer's point and have modified the text on p. 13 from "...where a central four-stranded parallel β -sheet interacts with the L5 β -hairpin" to "...where a central four-stranded parallel β -sheet sits in close proximity to the L5 β -hairpin". While the experiments proposed by the Reviewer (i.e. to truncate L5 in EstA and Ag43 to test if L5 has a conserved role in folding of passenger domains outside of the SPATE subfamily) presents a future direction of work that we will pursue, such analysis however, is beyond the scope of the current study. Since we already have an optimised pulse-chase assay for Pet and antibodies that specifically recognise the Pet passenger domain, we have added as Supplementary Fig. 8, pulse chase assays of Pet^{L5EstA} and Pet^{L5AIDA-I}, mutants where the residues in the Pet L5 β -hairpin were replaced with residues from the EstA and AIDA-I β -hairpins, respectively. We show that both the EstA and AIDA-I β -hairpins can act as nucleation sites for passenger folding.

13- Data on the autotransporter EspP lacking L5 have been published (Barnard et al., Nat Struct Mol Biol. 2007) and should be discussed in the context of the current work.

We appreciate the Reviewer's point and have modified the text on pp. 6-7 to make it clear that our findings upon truncation of L5 are consistent with data from a previous study on EspP where Barnard et al. (2007) aimed to investigate the contribution of L5 on β -barrel domain stability. In addition to finding that

truncation of L5 in EspP [EspP Δ 1(Δ L5); Δ E₁₂₃₈-E₁₂₅₃] had no effect on the stability of this domain, the authors also noted no effect on passenger domain translocation or cleavage. It is likely that Barnard et al. (2007) did not pick up on the defect in passenger folding upon truncation of L5 in EspP because they used a truncation mutant of native EspP [EspP Δ 1(Δ L5)] that contains 116 residues (i.e. ~13 kDa) of the passenger domain immediately N-terminal to the L5 truncated β -barrel domain in pulse-chase assays that were carried out in an *E. coli* strain lacking OmpT (AD202).

Minor comments:

1- line 71: remove space between (π - π)

We have made this change to the text.

2- line 117: “full-length Pet”: please make clear that this refers to the species without signal sequence in this case

We have modified the text on p. 6 to make it clear that in this study, “full-length” Pet refers to a Pet species minus its signal peptide.

3- Fig. 1e, right panel, -PK: please align ‘0, 1, 2, 5, 10 15 min’ annotation better with lanes. Confusing.

We have made this change to the figure.

4- Fig. 1g looks a bit sloppy, which disturbs interpretation. Much clearer pictures of the same assay were produced in previous publications by the authors.

We have addressed this concern as outlined in our response to Major comment no. 8.

5- Supp Fig. 5a: SDS-PAGE lanes Pet Δ L5 Δ 1-554 looks strange: half of the right hand lane seems to be missing

We have made this change to the figure such that the full lane is now visible.

6- Lines 445-450: concentration or % glucose and arabinose used should be specified.

The percentage of glucose and arabinose used in the pulse-chase assays are clearly stated in the ‘Reagents and bacterial strains’ section of the Methods, i.e. 0.2% (vol/vol) D-glucose, 0.02% (vol/vol) L-arabinose (see p. 21). We therefore don’t believe that it is necessary to restate the percentage of glucose and arabinose used in the ‘*In vivo* protein expression assay’ section of the Methods.

Reviewer #2 (Remarks to the Author):

Autotransporters are central virulence factors in the pathogenesis of many Gram-negative bacterial species. These proteins are composed of a translocator domain in the bacterial outer membrane and a passenger domain that is transported to the extracellular milieu. Folding of the passenger domain is a key event in secretion and activity. Folding occurs only once the passenger domain is exposed on the extracellular surface, but it remains unknown what triggers this reaction. In this work, Leyton and colleague discover that the region corresponding to the extracellular loop 5 of the translocator domain is essential for passenger domain folding. By using *in vivo* and *in vitro* experimental approaches they conclude that the β -hairpin conformation of loop 5 functions in nucleating the β -helical folding of the passenger domain.

This is a well written manuscript that will have a broad impact in the field of protein secretion. The authors show clear evidence that loop 5 is necessary for correct passenger domain folding, which is a novel function of the autotransporter translocator domain. The conclusion that its β -hairpin structure nucleates folding of the β -helical passenger domain remains an attractive hypothesis and not a proven fact, as it might appear from reading the abstract (see second last sentence) and other parts of the text. Even in the absence of such proof, this remains a relevant piece of work and it should be considered for publication once the authors address the following points.

We have modified the abstract and parts of the text with this in mind.

1) It appears odd that the authors compare the biogenesis of Pet in the BW25133 Δ OmpT and the Top10

strain. A direct comparison of pulse-chase Pet expression in BW25133 and BW25133 deltaompT should be shown.

We thank the Reviewer for picking up on this important point. We have addressed this concern as outlined in our response to Reviewer 1 (see Major comment no. 9).

2) Differently from what stated by the authors (page 7, Figure S1), deletion of loop4 delays of at least 5 min passenger domain cleavage and the appearance of its 70 kDa PK resistant fragment. Does loop 4 influence the activity of loop 5 in promoting correct passenger domain folding? In the absence of loop 5, may loop 4 be incorrectly positioned and interfere with the folding of the passenger domain. The authors should test the biogenesis of a Pet construct that lacks both loop 4 and loop 5.

We appreciate the Reviewer's point and have modified the text on p. 8 to make it clear that the mutant with a partially truncated L4 β -hairpin (renamed Pet Δ L4P) displays a concomitant delay in the arrival of the passenger domain at the cell surface and autocatalytic processing into distinct passenger and β -barrel domain fragments relative to that observed in wild type Pet. Furthermore, as outlined in our response to Reviewer 1 (see Major comment no. 2), complete removal of the L4 β -hairpin to create a new Pet Δ L4 results in a substantial reduction of passenger domain translocation as shown previously for the non-SPATE autotransporter BrkA (see Zhai et al., 2011 Biochem J) maybe coupled with a minor defect in folding. We hypothesise that L4 functions to restrain L5 movement and ensure that L5 is correctly positioned to interact with the passenger during translocation, thereby indirectly promoting folding of this domain. Or alternatively, that the folding of the passenger is compromised due to events that precede folding such as passenger translocation, which may occur non-optimally through the L4-truncated β -barrel domain, perhaps resulting in misfeeding of the passenger domain to L5. While we also constructed a double L4 and L5 mutant (see Fig. a below), we did not include the corresponding pulse-chase assays in the revised manuscript because the double mutant displayed a translocation and folding defect that is indistinguishable to that observed for the new Pet Δ L4 mutant (Fig. b below).

3) PetL5 β 1/G, PetL5OmpF, PetL β 1/P: Processing of the passenger domain cannot be detected. To which extent is the passenger domain secreted? Would the passenger domain accumulate in the BW25133 deltaompT strain? PK accessibility could be due to an improperly folded translocator domain?

We thank the Reviewer for picking up on this important point. We have addressed this concern as outlined in our response to Reviewer 1 (see Major comment no. 10).

4) What would be the effect of replacing Pet loop5 with an α -helical hairpin similar to that of FadL loop 3? This would be an interesting experiment, which might (or might not) support the folding nucleation hypothesis

The Reviewer proposes an interesting experiment along the lines of a series of experiments that we are pursuing in the lab. In line with the hypothesis formed in our manuscript, preliminary data from our lab that exactly addresses this show that this α -helical replacement in L5 severely hampers folding of the passenger, with signals similar to those observed for e.g. β 1 replacement to G (i.e. trace amounts of the 70 kDa fragment are visible after 40 min, consistent with our hypothesis). We choose not to include these data in the current manuscript, as it does not provide unambiguous proof for our hypothesis by itself. Clearly, a whole series of experiments would have to be initiated to substantiate that single finding. As the Reviewer points out, proving the hypothesis, which is based upon the data in this manuscript, will be very challenging. The lab is pursuing several avenues to achieve exactly that, but the body of work that will be required to give a convincing amount of evidence for it falls beyond the scope and timeframe for this work. We emphasize that excluding the α -helical replacement from this work does not hamper the interpretation of our results, or the formulation of the hypothesis; and, a priori, the α -helical replacement by itself would not provide sufficient evidence to prove the hypothesis neither.

5) Deletion of loop 5 does not affect the stability of the translocator domain. Why is the amount of the EspP β -barrel lacking loop5 lower than the amount of the wild type β -barrel domain in figure 3b? What is the overall level of expression/stability of the two constructs used in this figure?

We are not entirely sure why the amount of the EspP β -barrel lacking L5 is lower than the amount of the wild type EspP β -barrel in this figure. We see this also with the wild type and L5 mutant Pet constructs (see Fig. 1g). The amount of whole cell lysates and total membrane preparations were normalised as described in the Methods, which would suggest differences in expression levels between the wild type and L5 mutant constructs. Notably, the antibodies specific for the Pet β -barrel domain used in the current study were generated against the entire Pet β -barrel domain (see Leyton et al., 2014 Nat Commun). L5 is a long extracellular loop that is therefore likely to be highly immunogenic. Hence, an alternative explanation is that the L5 truncation mutants generate a weaker immunoblot signal because they lack this epitope. Importantly, these differences do not hamper the interpretation of our results, which are further validated through thermal melts of *in vitro* refolded β -barrels showing that the apparent thermal stability of the wild type β -barrel is indistinguishable from that of the L5-truncated β -barrel (Fig. 4g).

6) The authors use an elegant *in vitro* approach to prove the critical role of loop 5 in the folding of the passenger domain. Certainly this experiment is the strongest evidence in favor of the folding nucleation hypothesis. To further reinforce this point, and given that this *in vitro* approach is already set-up in their labs, the authors should investigate whether a short polypeptide corresponding to the loop5 beta hairpin has any *in trans* activity on the folding of the purified passenger domain.

We appreciate the Reviewer's point and while the experiments proposed by the reviewer present a future direction of work that we will pursue, such analysis however, is beyond the scope of the current study. This is because there would be a substantial amount of optimisation necessary. For example, what is the ideal peptide length? Will the peptide adopt the correct secondary structure in LDAO refolding buffer when not covalently attached to the β -barrel domain? Etc.

7) Would it be possible that loop 5 influences the correct positioning of the C-terminal region of the passenger domain, which has to be at a certain distance (or orientation) from the β -barrel domain in order to explore different possible conformations and acquire correct folding? Proving that the β -hairpin of loop 5 nucleates folding of the passenger domain will be extremely challenging. This is currently the main hypothesis discussed by the authors. Other alternatives concerning the molecular role of loop 5 need to be considered.

We appreciate the Reviewer's point. However, if L5 simply partitions the C-terminal region of the passenger domain so that it has more space available to acquire its native fold (as we understand the Reviewer suggests), we struggle to rationalise why the presence of L5 accelerates passenger folding in solution when several groups already reported that unassisted passenger folding is slow. However, we have now modified the abstract and sections of the text to make it clearer that while our data suggest that L5 mediates passenger folding through β -strand augmentation, that this mechanism is not proven, but rather, is our current hypothesis.

Reviewer #3 (Remarks to the Author):

This is a very clearly written manuscript on passenger domain folding in autotransporters, and specifically on the influence of an outer membrane surface loop (L5) on this complex protein folding and transport process. I have some suggestions on how to improve the manuscript, and I do think that additional controls are necessary to be able to generalize claims about this - clearly important - loop region.

We thank the Reviewer for picking up this important point. We have now included all of the additional control experiments requested as detailed below.

- to put the work in perspective, I would suggest to mention (maybe in the abstract and certainly in the introduction) that there are different classes of autotransporters, and that the work presented here is only done on one of them - the "classical" autotransporters, of type Va secretion systems.

We have modified the text in the introduction (pp. 2-4) to make it clear that our work and therefore conclusions are focused on the classical autotransporters.

Other autotransporter have different barrel features, including oligomerization, or the barrels being at the N-terminus. The current phrasing does not clarify this, but rather makes it sound as if the features described in this manuscript are universal (e.g. the barrel being at the C-terminus...). All of this is probably included in the citations used by the authors, but should be mentioned explicitly for the non-expert reader.

We have modified the text in the introduction (p. 3) to make it clear that the N-terminal and C-terminal domain organisation described in the text relates to the classical autotransporters.

This is also relevant for the discussion part where the authors generalize the importance of the effects found here, but make it sound as if this is true for all autotransporters (instead of one class)

The majority of the discussion is in the context of the Pet or SPATE autotransporters. Through modifications in the introduction, as discussed above, we now clarify that our findings relate to the classical autotransporters. We hope that the Reviewer now agrees that we have better placed the interpretation of the results.

Regarding the experimental work, I have a few comments that might need addressing:

- several of the blots are of poor quality. Specifically figure 1g is very smeary, and figure 4c is totally overloaded in 2 of the lanes.

We appreciate the Reviewer's point and have addressed the concern about Fig. 1g as outlined in our response to Reviewer 1 (see Major comment no. 8). As for Fig. 4c, while the lanes containing undigested samples (i.e. - trypsin) appear overloaded, the same amount of protein was run in these lanes and in the lanes containing digested samples (i.e. + trypsin). The signal in the lanes containing undigested samples appears stronger because the passenger domain is intact in contrast to the signal in the lanes containing digested samples, which is weaker because the passenger domain has been digested into a smaller trypsin-resistant fragments.

- I am missing an (obvious?) control in the experiments: a control for periplasmic protection in the PK digestion. The experiments with the ompT deletion mutant suggest that PetdeltaL5 is surface exposed, but to rule out periplasmic degradation by PK (due to loss of OM integrity in the mutant), the authors should show a blot of a periplasmic protein (maybe SurA or similar, or something like MBP? The latter can be expressed easily in E.coli and exists in variants of different stability) that is unaffected by PK in both the WT and the mutant.

We thank the Reviewer for picking up on this point. We have addressed this concern as outlined in our response to Reviewer 1 (see Major comment no. 6).

- to make a general claim that L5 is "the nucleation site for passenger folding" in other types of ATs as well (and again: only the type Va ones?), maybe this should be tested directly, e.g. by replacing the loop in their system with one from a non-SPATE autotransporter (similar to the FadL replacement shown here)?

We thank the Reviewer for picking up on this point. We have addressed this concern as outlined in our response to Reviewer 1 (see Major comment no. 12).

- Why is there no signal for the passenger in the trypsinated refolding experiments at time point 0? Should there not be a clear signal? This needs some discussion I think?

In this assay, the time points refer to the refolding time as opposed to the length of trypsin treatment (we have modified the labels in Fig. 4c to make this more clear). This unique tryptic fingerprint is caused by the transient exposure of trypsin-accessible Arg and Lys residues during the folding of the passenger until time = 120 min where the majority of the passenger molecules have folded into a largely protease-resistant structure. As such, it reasons that there is no trypsin-resistant fragment at time = 0 min because the passenger molecules have not yet had enough time to fold into their native conformations.

REVIEWERS' COMMENTS:

Reviewer #1 (Remarks to the Author):

Yuan and co-workers addressed the majority of the points raised by this referee during review of the original manuscript adequately. Yet there are some outstanding issues that the authors should address. Please find below my response to the concerning answers by the authors:

Original comment:

1- line 93,94: The crystal structure of EspP was used for topology modeling of the Pet beta-domain. Was the modeling done 'by hand' or aided by software? Please provide details about the software used and the underlying considerations. Why does the topology model not follow the cited EspP barrel crystal structure (pdb: 3slj), which shows that L4 and L5 are not independent loops but together form a 4-stranded betasheet? It seems more appropriate to align L4-L5 such that polar contacts between these loops that are visible in the pdb file are represented.

Authors:

Topology modelling of the Pet β -barrel domain was done by hand based on the boundaries defined by the crystal structure of the EspP β -barrel domain (PDB code 3SLJ). The fact that the β -barrel domains of both proteins are 277 amino acid residues in length and that these β -barrels share ~91% sequence identity rendered the process of manual topology modelling simple and straightforward. Our topology model does not show that L4 and L5 form a 4-stranded β -sheet as shown in the crystal structure of the EspP β -barrel domain with PDB code 3SLJ because these loops are mobile and therefore not always in this conformation. For example, the original crystal structure of the EspP β -barrel domain (PDB 2QOM) shows L5 folded into the pore of the β -barrel. Since this crystal structure shows the β barrel postcleavage and release of the passenger, the authors (Barnard et al., 2007 NSMB) proposed that L5 in this 'closed' conformation restricts solvent access into the bacterium from the extracellular side. In this crystal structure, L3 and L4 remain partially unresolved. In contrast, the EspP β -barrel domain cited in our work (PDB code 3SLJ) shows a pre-cleavage structure (i.e. the most extreme C-terminal portion of the passenger domain is still attached) where L5 is in the 'open' conformation, projecting into the extracellular space. The authors (Barnard et al., 2012 J Mol Biol) show that the uncleaved passenger domain, which traverses the length of the pore, also projecting into the extracellular space, physically blocks L5 from making extensive contact with the β -barrel. Since passenger domain translocation precedes cleavage (Skillman et al., 2005 Mol Microbiol) and folding of the passenger domain is proposed to be vectorial (Braselman & Clark, 2012 J Phys Chem Lett), we used the EspP β -barrel domain with PDB code 3SLJ in our work because it better represents the conformation of L5 during translocation. Nevertheless, given that L3, L4 and L5 are mobile and the equilibrium solution conformations are unknown, we prefer not to align L4 and L5 such that the polar contacts between these two loops are visible.

Response Reviewer 1:

Fair points. Yet, it would assist the readership if the bottom-line of this rationale (selected structure represents translocation state and topology model does not strictly follow pdb 3SLJ to reflect mobility of L3, L4 and L5 observed in other pdb's and studies) is included in the manuscript, either in the results or the legend to Figure 1.

Original comment:

2- Based on the Pet deletion mutant tested in Supp Fig. 1a/b the authors draw the conclusion that L4 is not critical for passenger folding, but only the tip of the loop has been removed, with half of the beta-hairpin still intact. In contrast, to study the role of L5, the complete loop beta-hairpin was deleted. What is the rationale behind deleting only part of L4? The remainder of L4 might still play a

role in passenger folding, either directly or indirectly. An L4 mutant lacking at least the complete beta-hairpin should be tested.

Authors:

All three Pet loop deletion mutants were designed by co-author Ian Henderson in 2006 based on the crystal structure of the NalP β -barrel domain (1UYN), which was published in 2004 (Oomen et al., 2004 EMBO). This occurred prior to publication of the NSMB paper by Barnard and co-authors reporting the crystal structure of the EspP β -barrel domain in 2007. Since the β -barrel domains of Pet and NalP only share $\sim 18\%$ sequence identity, the predicted boundaries of some of the Pet loops were not precise. Nevertheless, we agree with the Reviewer's point and have added to Supplementary Fig. 2, pulse-chase assays of a new L4 truncation mutant in *E. coli* TOP10 showing that complete removal of the L4 β -hairpin results in a substantial reduction of passenger domain translocation as shown previously for autotransporter BrkA (see Zhai et al., 2011 Biochem J). Pulse-chase expression of the full L4 truncation mutant in *E. coli* BW25113 Δ ompT showed an increase in cleavage efficiency, suggesting that part of the passenger population is unfolded. Since our original data using a L4 mutant with a partially deleted β -hairpin shows a minor translocation defect, but no obvious folding defect, these data suggest that L4 mainly acts during translocation and that if L4 has a role in passenger folding, its role during folding is non-essential. We have revised the Discussion with these new data in mind.

Reviewer 1:

I agree with the authors that the primary function of L4 seems to lie in the translocation of the passenger across the membrane. However, the conclusion that L4 is not essential for the folding process, still only applies to the part of L4 removed in mutant Δ L4P. In fact, in the above rebuttle the authors state that "Pulse-chase expression of the full L4 truncation mutant in *E. coli* BW25113 Δ ompT showed an increase in cleavage efficiency, suggesting that part of the passenger population is unfolded". Hence, despite the presence of an intact L5, passenger folding was disturbed arguing for a role of L4 in this process, either directly or indirectly. Indeed, the authors speculate about an indirect role for L4 in the model proposed in the Discussion and are advised to allude to this already when interpreting the experiments with the Δ L4 mutant in the Results.

Original comment:

Length of the L5 beta-hairpin as a critical parameter is based on the observation that a 4 aa hairpin of OmpF does not mediate passenger folding as opposed to the longer hairpins of Pet and FadL. Rather than using a heterologous OmpF hairpin (that seems to interfere with upstream processes; see comment above), constructs carrying shortened versions of the endogenous Pet L5 hairpin should be used to make this point.

Authors:

We thank the Reviewer for this suggestion. We have added as Supplementary Fig. 6, pulse-chase assays of a series of L5 truncation mutants, which show that the endogenous β -hairpin must be of a certain length (comprising of no less than five residues) to scaffold folding of the Pet passenger domain.

Response Reviewer 1:

The set of data obtained with a series of new L5 truncation mutants addresses this specific concern rather adequately. However, the conclusion that a minimum hairpin of length of 5aa is required to sustain passenger folding, contradicts the finding that a 3aa hairpin of AIDA-I can also restore this process. The authors point this out themselves at p.14 and present a somewhat lengthy discussion on potential explanations to reconcile these findings. This raises the questions about the value of the information presented in Fig. S6 and the conclusion drawn. As the results are not pertinent to the

essence of the paper it may be better to remove the part addressing the relationship between length and functionality of L5.

Original comment:

6- Lines 445-450: concentration or % glucose and arabinose used should be specified.

Authors:

The percentage of glucose and arabinose used in the pulse-chase assays are clearly stated in the 'Reagents and bacterial strains' section of the Methods, i.e. 0.2% (vol/vol) D-glucose, 0.02% (vol/vol) L-arabinose (see p. 21). We therefore don't believe that it is necessary to restate the percentage of glucose and arabinose used in the 'In vivo protein expression assay' section of the Methods.

Response Reviewer 1:

Apologies for missing that. Yet, it would be clearer for the reader if the percentages are stated with the description of the expression assay itself. Also, since the description reads "cells were resuspended in arabinose" it was not entirely clear to me that this means medium supplemented with arabinose as stated in the Reagents section. Are cells actually resuspended in plain arabinose or in medium with arabinose? Please clarify to assist the reader.

Reviewer #2 (Remarks to the Author):

In this revised version, the authors have addressed the reviewer's comments, adding requested controls and improving the overall quality of their manuscript. This manuscript will be an important reference point for future studies.

I have a couple of additional points that I suggest the authors to address.

1) Figure S4: the mutation L5beta1/G does not affect the onset of passenger secretion (as judged by looking at the amount of full-length protein that is accessible to PK). Nevertheless, completion of passenger domain secretion is clearly delayed. In fact, the relative amount of processed passenger domain (106 kDa fragment compared to the full-length 136 kDa fragment) that accumulates after 40 min in the absence of PK is similar to the relative amount of passenger domain formed from wild type Pet after only 10 min. This result indicates a delay of 30 min to complete secretion for the mutant Pet. This delay must be properly described in the text, lines 206-215. The authors may argue that in the construct harboring the L5beta1/G mutation, the amount of fully secreted passenger domain is delayed probably due to its impaired folding, which is a driving force to complete secretion. The other constructs tested in Figure S4 show similar delays in secretion that the authors should comment on.

2) Folding nucleation is a reasonable working model and not a demonstrated feature. The revised version of this manuscript does not provide further evidence strengthening that loop 5 nucleates/templates folding of the passenger domain. The authors should change the following points in the text.

Line 322, change "have the potential to be nucleation sites for passenger folding" to "have the potential to promote folding".

Line 348, "To test if folding of the passenger domain occurs through the same nucleation mechanism in the absence of the cellular machinery..." to "To investigate if Pet loop 5 is critical for folding of the passenger domain independently of the cellular machinery..."

Line 380, "L5 serves as a structural template for the nucleation of passenger folding" to "L5 assists folding possibly by providing a structural template for the nucleation of passenger folding"

Reviewer #3 (Remarks to the Author):

For reference, I was (and still am) referee 3... all of my comments have been addressed satisfactorily except for one. Admittedly this may be a misunderstanding on my side but I think clarification is needed both in the response to referees, and in the actual manuscript text:

My original question was: "Why is there no signal for the passenger in the trypsinated refolding experiments at time point 0? Should there not be a clear signal? This needs some discussion I think?" The authors answer was: "In this assay, the time points refer to the refolding time as opposed to the length of trypsin treatment (we have modified the labels in Fig. 4c to make this more clear). This unique tryptic fingerprint is caused by the transient exposure of trypsin-accessible Arg and Lys residues during the folding of the passenger until time = 120 min where the majority of the passenger molecules have folded into a largely protease-resistant structure. As such, it reasons that there is no trypsin-resistant fragment at time = 0 min because the passenger molecules have not yet had enough time to fold into their native conformations."

Are you saying that the passenger is only seen in the blot when trypsinated (or can only be seen by the antibody if fully folded)? What I meant to say in my original question is: where is the full-length (undigested) band that should be picked up with an antibody directed to the passenger at all times? Please mark clearly in figure 4c top panel(s) which band corresponds to the undigested autotransporter. If this is not visible, please explain why. Or did you mean to say that at time point 0 it is not yet expressed (you suggest 'not yet folded' in your response)?

I am also missing an explanation of why in the panel labeled "petdeltaL5delta1-554" all lanes look like there is almost nothing in them (with the passenger antibody)? Again, please label the expected bands (full-length and characteristic digests). It might be a matter of figure/PDF quality but in this second panel I'm not seeing anything really.

Point-by-point rebuttal for NCOMMS-17-19249-A

We thank the Reviewers for their positive appraisals of the revised paper and their recommendations for publication in Nature Communications. We noted misinterpretation of our results by the Reviewers in a couple of instances, but for the most part, we found their comments to be constructive and have addressed the issues raised with revision to the text (using the ‘tracked changes’ feature), as detailed point-by-point below:

REVIEWERS' COMMENTS:

Reviewer #1 (Remarks to the Author):

Yuan and co-workers addressed the majority of the points raised by this referee during review of the original manuscript adequately. Yet there are some outstanding issues that the authors should address. Please find below my response to the concerning answers by the authors:

Original comment:

1- line 93,94: The crystal structure of EspP was used for topology modeling of the Pet beta-domain. Was the modeling done ‘by hand’ or aided by software? Please provide details about the software used and the underlying considerations. Why does the topology model not follow the cited EspP barrel crystal structure (pdb: 3slj), which shows that L4 and L5 are not independent loops but together form a 4-stranded betasheet? It seems more appropriate to align L4-L5 such that polar contacts between these loops that are visible in the pdb file are represented.

Authors:

Topology modelling of the Pet β -barrel domain was done by hand based on the boundaries defined by the crystal structure of the EspP β -barrel domain (PDB code 3SLJ). The fact that the β -barrel domains of both proteins are 277 amino acid residues in length and that these β -barrels share ~91% sequence identity rendered the process of manual topology modelling simple and straightforward. Our topology model does not show that L4 and L5 form a 4-stranded β -sheet as shown in the crystal structure of the EspP β -barrel domain with PDB code 3SLJ because these loops are mobile and therefore not always in this conformation. For example, the original crystal structure of the EspP β -barrel domain (PDB 2QOM) shows L5 folded into the pore of the β -barrel. Since this crystal structure shows the β barrel postcleavage and release of the passenger, the authors (Barnard et al., 2007 NSMB) proposed that L5 in this ‘closed’ conformation restricts solvent access into the bacterium from the extracellular side. In this crystal structure, L3 and L4 remain partially unresolved. In contrast, the EspP β -barrel domain cited in our work (PDB code 3SLJ) shows a pre-cleavage structure (i.e. the most extreme C-terminal portion of the passenger domain is still attached) where L5 is in the ‘open’ conformation, projecting into the extracellular space. The authors (Barnard et al., 2012 J Mol Biol) show that the uncleaved passenger domain, which traverses the length of the pore, also projecting into the extracellular space, physically blocks L5 from making extensive contact with the β -barrel. Since passenger domain translocation precedes cleavage (Skillman et al., 2005 Mol Microbiol) and folding of the passenger domain is proposed to be vectorial (Braselman & Clark, 2012 J Phys Chem Lett), we used the EspP β -barrel domain with PDB code 3SLJ in our work because it better represents the conformation of L5 during translocation. Nevertheless, given that L3, L4 and L5 are mobile and the equilibrium solution conformations are unknown, we prefer not to align L4 and L5 such that the polar contacts between these two loops are visible.

Response Reviewer 1:

Fair points. Yet, it would assist the readership if the bottom-line of this rationale (selected structure represents translocation state and topology model does not strictly follow pdb 3SLJ to reflect mobility of L3, L4 and L5 observed in other pdb’s and studies) is included in the manuscript, either in the results or the legend to Figure 1.

We have made this change to the text in the Legend to Figure 1 (p. 28).

Original comment:

2- Based on the Pet deletion mutant tested in Supp Fig. 1a/b the authors draw the conclusion that L4 is not critical for passenger folding, but only the tip of the loop has been removed, with half of the beta-hairpin still intact. In contrast, to study the role of L5, the complete loop beta-hairpin was deleted. What is the rationale behind deleting only part of L4? The remainder of L4 might still play a role in passenger folding, either directly or indirectly. An L4 mutant lacking at least the complete beta-hairpin should be tested.

Authors:

All three Pet loop deletion mutants were designed by co-author Ian Henderson in 2006 based on the crystal structure of the NalP β -barrel domain (1UYN), which was published in 2004 (Oomen et al., 2004 EMBO). This occurred prior to publication of the NSMB paper by Barnard and co-authors reporting the crystal structure of the EspP β -barrel domain in 2007. Since the β -barrel domains of Pet and NalP only share ~18% sequence identity, the predicted boundaries of some of the Pet loops were not precise. Nevertheless, we agree with the Reviewer's point and have added to Supplementary Fig. 2, pulse-chase assays of a new L4 truncation mutant in *E. coli* TOP10 showing that complete removal of the L4 β -hairpin results in a substantial reduction of passenger domain translocation as shown previously for autotransporter BrkA (see Zhai et al., 2011 Biochem J). Pulse-chase expression of the full L4 truncation mutant in *E. coli* BW25113 Δ ompT showed an increase in cleavage efficiency, suggesting that part of the passenger population is unfolded. Since our original data using a L4 mutant with a partially deleted β -hairpin shows a minor translocation defect, but no obvious folding defect, these data suggest that L4 mainly acts during translocation and that if L4 has a role in passenger folding, its role during folding is non-essential. We have revised the Discussion with these new data in mind.

Reviewer 1:

I agree with the authors that the primary function of L4 seems to lie in the translocation of the passenger across the membrane. However, the conclusion that L4 is not essential for the folding process, still only applies to the part of L4 removed in mutant Δ L4P. In fact, in the above rebuttle the authors state that "Pulse-chase expression of the full L4 truncation mutant in *E. coli* BW25113 Δ ompT showed an increase in cleavage efficiency, suggesting that part of the passenger population is unfolded". Hence, despite the presence of an intact L5, passenger folding was disturbed arguing for a role of L4 in this process, either directly or indirectly. Indeed, the authors speculate about an indirect role for L4 in the model proposed in the Discussion and are advised to allude to this already when interpreting the experiments with the Δ L4 mutant in the Results.

We have made this change to the text (p. 8).

Original comment:

Length of the L5 beta-hairpin as a critical parameter is based on the observation that a 4 aa hairpin of OmpF does not mediate passenger folding as opposed to the longer hairpins of Pet and FadL. Rather than using a heterologous OmpF hairpin (that seems to interfere with upstream processes; see comment above), constructs carrying shortened versions of the endogenous Pet L5 hairpin should be used to make this point.

Authors:

We thank the Reviewer for this suggestion. We have added as Supplementary Fig. 6, pulse-chase assays of a series of L5 truncation mutants, which show that the endogenous β -hairpin must be of a certain length (comprising of no less than five residues) to scaffold folding of the Pet passenger domain.

Response Reviewer 1:

The set of data obtained with a series of new L5 truncation mutants addresses this specific concern rather adequately. However, the conclusion that a minimum hairpin of length of 5aa is required to sustain passenger folding, contradicts the finding that a 3aa hairpin of AIDA-I can also restore this process. The authors point this out themselves at p.14 and present a somewhat lengthy discussion on potential explanations to reconcile these findings. This raises the questions about the value of the information

presented in Fig. S6 and the conclusion drawn. As the results are not pertinent to the essence of the paper it may be better to remove the part addressing the relationship between length and functionality of L5.

We agree with the Reviewer and have removed the section on the minimum length of the Pet β -hairpin that is required to template passenger folding (p. 11) and Supplementary Fig. 6. We also removed the interpretation of these results based on the AIDA-I findings (p. 14) and have included a sentence stating that “While it is unclear why the first β -strand in the AIDA-I β -hairpin, but not the OmpF β -hairpin, is able to promote folding of the Pet passenger domain, despite both β -strands being 4-residues in length, it supports the notion that β -strand propensity and length might play a role in passenger folding” (pp. 13-14). We speculate that the propensity and length of the β -hairpin are somehow in an act of balance: a strong β -hairpin might require less in length, while a β -hairpin with low propensity might require a longer β -hairpin. Since we can’t prove that Pet, EspP, AIDA-I, and FadL have higher β -hairpin propensities than OmpF in the context of the Pet structure, we have omitted this hypothesis from the manuscript. Nevertheless, the ambiguity in the minimum length of a β -hairpin (endogenous versus heterologous) that is required to template folding of the Pet passenger domain presents a future direction of work that we will pursue.

Original comment:

6- Lines 445-450: concentration or % glucose and arabinose used should be specified.

Authors:

The percentage of glucose and arabinose used in the pulse-chase assays are clearly stated in the ‘Reagents and bacterial strains’ section of the Methods, i.e. 0.2% (vol/vol) D-glucose, 0.02% (vol/vol) L-arabinose (see p. 21). We therefore don’t believe that it is necessary to restate the percentage of glucose and arabinose used in the ‘In vivo protein expression assay’ section of the Methods.

Response Reviewer 1:

Apologies for missing that. Yet, it would be clearer for the reader if the percentages are stated with the description of the expression assay itself. Also, since the description reads “cells were resuspended in arabinose” it was not entirely clear to me that this means medium supplemented with arabinose as stated in the Reagents section. Are cells actually resuspended in plain arabinose or in medium with arabinose? Please clarify to assist the reader.

The percentage of glucose and arabinose used in the pulse-chase assays are now also stated in the “In vivo protein expression assay’ section of the Methods (p. 21). We have also made it clear that in this assay, it is the medium that is supplemented with either glucose or arabinose.

Reviewer #2 (Remarks to the Author):

In this revised version, the authors have addressed the reviewer’s comments, adding requested controls and improving the overall quality of their manuscript. This manuscript will be an important reference point for future studies.

I have a couple of additional points that I suggest the authors to address.

1) Figure S4: the mutation L5beta1/G does not affect the onset of passenger secretion (as judged by looking at the amount of full-length protein that is accessible to PK). Nevertheless, completion of passenger domain secretion is clearly delayed. In fact, the relative amount of processed passenger domain (106 kDa fragment compared to the full-length 136 kDa fragment) that accumulates after 40 min in the absence of PK is similar to the relative amount of passenger domain formed from wild type Pet after only 10 min. This result indicates a delay of 30 min to complete secretion for the mutant Pet. This delay must be properly described in the text, lines 206-215. The authors may argue that in the construct harboring the L5beta1/G mutation, the amount of fully secreted passenger domain is delayed probably due to its impaired folding, which is a

driving force to complete secretion. The other constructs tested in Figure S4 show similar delays in secretion that the authors should comment on.

As explained on p. 9, our data suggest that the reason the mature 106 kDa species of PetL5 β 1/G cannot be detected in the TOP10 strain until the 40 min time point is because the passenger is autocatalytically cleaved from its β -barrel domain, yet is unable to efficiently fold into its native conformation and is therefore susceptible to degradation by OmpT. Pulse-chase expression of PetL5 β 1/G in BW25113 Δ ompT showed that this mutant is not delayed in secretion (i.e. passenger translocation + autocatalysis). For example, like wild type Pet, the passenger domain of PetL5 β 1/G and PetL5OmpF were largely surface exposed by the 10 min time point judging by the susceptibility of the 136 kDa full-length species to proteinase K (Supplementary Fig. 4, right panels, + PK). Furthermore, autocatalysis of Pet, PetL5 β 1/G, and PetL5OmpF passenger domains was also evident by the 10 min time point (Supplementary Fig. 4, left panels, - PK). Only PetL5 β 1/P showed a slight (5 min) delay in autocatalysis (Supplementary Fig. 4, left panel, - PK), but not translocation (Supplementary Fig. 4, right panel, + PK). For reasons unknown, cleavage of all three mutants appeared to occur with reduced efficiency relative to wild type, with only ~50-70% processing of the 136 kDa full-length species to the 106 kDa mature species after 40 min (Supplementary Fig. 4, left panels, - PK). Importantly, proteinase K digestion of Pet into a ~70 kDa protease-resistant fragment at the 10 min and 15 min time points (Supplementary Fig. 4, right panel, + PK) shows that the presence of both the 136 kDa and 106 kDa species at these time points (Supplementary Fig. 4, left panel, - PK) does not interfere with the generation of the ~70 kDa protease-resistant fragment, which is a marker for natively folded passenger domain molecules. From this we can extrapolate with confidence that the reduced cleavage efficiency observed in PetL5 β 1/G, PetL5 β 1/P, and PetL5OmpF (Supplementary Fig. 4, left panels, - PK) is not responsible for the defect in passenger folding exhibited by these mutants. With this in mind and because the reduced cleavage efficiency of these mutants is not pertinent to the essence of the paper, we don't feel it is necessary to comment on this phenotype in the manuscript.

2) Folding nucleation is a reasonable working model and not a demonstrated feature. The revised version of this manuscript does not provide further evidence strengthening that loop 5 nucleates/templates folding of the passenger domain. The authors should change the following points in the text.

Line 322, change “have the potential to be nucleation sites for passenger folding” to “have the potential to promote folding”.

We have made this change to the text (p. 13).

Line 348, “To test if folding of the passenger domain occurs through the same nucleation mechanism in the absence of the cellular machinery...” to “To investigate if Pet loop 5 is critical for folding of the passenger domain independently of the cellular machinery...”

We have made this change to the text (p. 14).

Line 380, “L5 serves as a structural template for the nucleation of passenger folding” to “L5 assists folding possibly by providing a structural template for the nucleation of passenger folding”

We have made this change to the text (p. 15).

Reviewer #3 (Remarks to the Author):

For reference, I was (and still am) referee 3... all of my comments have been addressed satisfactorily except for one. Admittedly this may be a misunderstanding on my side but I think clarification is needed both in the response to referees, and in the actual manuscript text:

For reasons outlined below, we believe that the Reviewer has misunderstood the basis of the trypsin assay and the interpretation of our results. Therefore, we have addressed the Reviewer's questions in our responses below, but we don't feel that it is necessary to adjust the manuscript text other than to include the following text in the Legend to Fig. 4: "In this assay, folding was initiated and aliquots of the reaction mixture, at increasing incubation times, were treated with trypsin and then mixed with SDS to stop folding" (p. 30). In this way it will be clear that the time points refer to the refolding time as opposed to the length of trypsin treatment, which could cause confusion in the reader otherwise.

My original question was: "Why is there no signal for the passenger in the trypsinated refolding experiments at time point 0? Should there not be a clear signal? This needs some discussion I think?"

The authors answer was: "In this assay, the time points refer to the refolding time as opposed to the length of trypsin treatment (we have modified the labels in Fig. 4c to make this more clear). This unique tryptic fingerprint is caused by the transient exposure of trypsin-accessible Arg and Lys residues during the folding of the passenger until time = 120 min where the majority of the passenger molecules have folded into a largely protease-resistant structure. As such, it reasons that there is no trypsin-resistant fragment at time = 0 min because the passenger molecules have not yet had enough time to fold into their native conformations."

Are you saying that the passenger is only seen in the blot when trypsinated (or can only be seen by the antibody if fully folded)? What I meant to say in my original question is: where is the full-length (undigested) band that should be picked up with an antibody directed to the passenger at all times? Please mark clearly in figure 4c top panel(s) which band corresponds to the undigested autotransporter. If this is not visible, please explain why. Or did you mean to say that at time point 0 it is not yet expressed (you suggest 'not yet folded' in your response?)?

As explained on pp. 14-15, folded Pet Δ 1-554 passenger (secreted by the wild type β -barrel) is largely resistant to trypsinolysis, generating trypsin-resistant fragments between ~35 and ~40 kDa, whereas the Pet Δ L5 Δ 1-554 passenger (secreted by the L5-truncated β -barrel) is degraded completely. The trypsin-resistant fragments between ~35 and ~40 kDa are only detected by the anti-passenger domain antibody when trypsin is present and when the passenger molecules have folded into their native conformations (i.e. a largely protease-resistant structure) (Fig. 4c, top panel, Pet Δ 1-554, + trypsin, 10-240 min). Our data suggest that there is no trypsin-resistant fragment at time = 0-5 min (Fig. 4c, top panel, Pet Δ 1-554, + trypsin, 0-5 min) because the passenger molecules have not yet had enough time to fold into their native conformations. Since in our assay (i) trypsin digests folded passenger molecules into trypsin-resistant fragments between ~35 and ~40 kDa, (ii) trypsin completely degrades unfolded passenger molecules and, (iii) the refolding reaction is only ~50% efficient, the 81 kDa full-length Pet Δ 1-554 and Pet Δ L5 Δ 1-554 species are only visible when the samples have not been treated with trypsin (Fig. 4c, top panel, Pet Δ 1-554 and Pet Δ L5 Δ 1-554, - trypsin, 240 min). Here, the ~51 kDa cleaved (i.e. mature) species are also visible since there is no trypsin to digest the passenger molecules into trypsin-resistant fragments between ~35 and ~40 kDa if folded or to completely degrade the passenger molecules if unfolded (Fig. 4c, top panel, Pet Δ 1-554 and Pet Δ L5 Δ 1-554, - trypsin, 240 min). Both the ~81 kDa and ~51 kDa species are already clearly indicated in Fig. 4c along with the trypsin-resistant fragments between ~35 and ~40 kDa (top panel, right hand side). Of note, our data suggest that a band corresponding to the cleaved β -barrels of Pet Δ 1-554 and Pet Δ L5 Δ 1-554 are visible at 5 and 0 min, respectively (Fig. 4c, bottom panel, Pet Δ 1-554 and Pet Δ L5 Δ 1-554, + trypsin, 5/0-240 min and - trypsin, 240 min) because, in the case of Pet Δ L5 Δ 1-554, this domain folds within the dead time of the experiment and in both cases, the folding of the Pet Δ 1-554 and Pet Δ L5 Δ 1-554 β -barrels is critical for the folding of the passenger domain. In other words, to assist folding of the passenger domain, the β -barrel must fold first.

I am also missing an explanation of why in the panel labeled "petdeltaL5delta1-554" all lanes look like there is almost nothing in them (with the passenger antibody)? Again, please label the expected bands (full-length and characteristic digests). It might be a matter of figure/PDF quality but in this second panel I'm not seeing anything really.

As explained above and on pp. 14-15, while the folded Pet Δ 1-554 passenger (secreted by the wild type β -barrel) is largely resistant to trypsinolysis, generating trypsin-resistant fragments between ~35 and ~40 kDa,

the Pet Δ L5 Δ 1-554 passenger (secreted by the L5-truncated β -barrel) is degraded completely. This is why there are no bands detected by the anti-passenger domain antibody in samples treated with trypsin (Fig. 4c, top panel, Pet Δ L5 Δ 1-554, + trypsin, 0-240 min). Therefore, it reasons that the 81 kDa full-length and 51 kDa mature Pet Δ 1-554 and Pet Δ L5 Δ 1-554 species are only visible when the samples have not been treated with trypsin (Fig. 4c, top panel, Pet Δ 1-554 and Pet Δ L5 Δ 1-554, - trypsin, 240 min).